# Mechanisms of Multimodal Synchronization: Insights from Decoder-Based Video-Text-to-Speech Synthesis

## Abstract

Unified decoder-only transformers have shown promise for multimodal generation, yet the mechanisms by which they synchronize modalities with heterogeneous sampling rates remain underexplored. We investigate these mechanisms through video-text-to-speech (VTTS) synthesis—a controlled task requiring fine-grained temporal alignment between sparse text, video, and continuous speech. Using a unified decoder-only transformer, dubbed `Visatronic`, trained on VoxCeleb2, we study: (i) how modalities contribute complementary information, (ii) how positional encoding strategies enable synchronization across heterogeneous rates, (iii) how modality ordering shapes the trade-off between in-domain performance and cross-domain transfer, (iv) how phoneme-level synchronization metrics provide diagnostic insight into per-phoneme timing errors. Our findings reveal that both "global sequential indexing" (unique position IDs across modalities) and "co-temporal ordered indexing" (identical IDs for temporally corresponding tokens) achieve strong synchronization performance, with co-temporal ordered indexing providing a simple mechanism without explicit timestamp metadata. Both text and video contribute complementary signals: text ensures intelligibility while video provides temporal cues and emotional expressiveness. Modality ordering reveals a consistent trade-off: video-first ordering achieves stronger in-domain performance while text-first ordering generalizes more robustly to unseen domains. Our findings also reveal, that diverse large-scale training enables transferable synchronization strategies. To enable fine-grained analysis, we also introduce `TimeSync`, a phoneme-level metric that reveals temporal misalignments overlooked by frame-level metrics. These insights establish VTTS as a valuable testbed for understanding temporal synchronization in unified multimodal decoders. Generated speech results are attached in the supplementary.

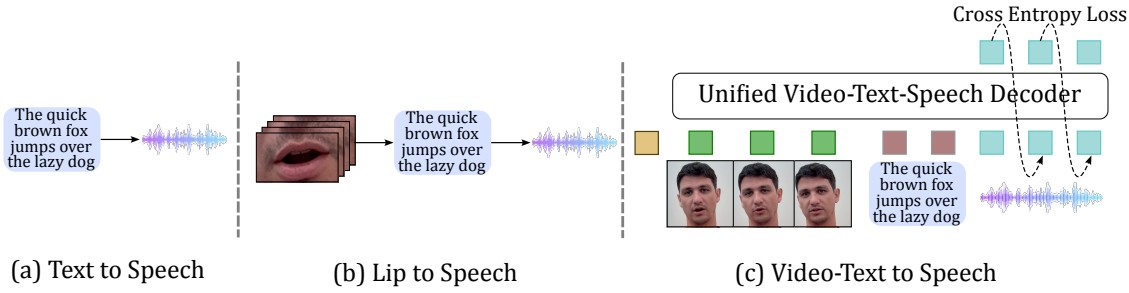

(a) Text to Speech   (b) Lip to Speech   (c) Video-Text to Speech

Figure 1: **`Visatronic` overview.** We study the video-text to speech (VTTS) task as a controlled playground to understand synchronization mechanisms in unified multimodal decoders. `Visatronic` is a unified decoder-only transformer that processes speaker tokens (yellow), video **v** (green), text **t** (red), and speech **s** (cyan) as discrete tokens in a shared sequence. We use this same color convention throughout the paper. The model is trained with cross-entropy loss $\mathcal{L}_{CE}$ to predict speech tokens, learning cross-modal interactions and temporal alignment across modalities with heterogeneous sampling rates (sparse text, 25fps video, 40Hz speech). By reducing confounding factors such as mixture-of-experts architectures and multi-stage training, VTTS enables systematic investigation of how unified decoders align temporal information.

# 1 Introduction

Unified decoder-only architectures have emerged as a powerful paradigm for multimodal generative tasks, demonstrating scalability across modalities (Kondratyuk et al., 2024; Yu et al., 2022; McKinzie et al., 2024; Team et al., 2024; Achiam et al., 2023; Zhan et al., 2024; Lu et al., 2024). Despite the empirical success of adapting pretrained large language models (LLMs) to multimodal inputs (Xu et al., 2025b; Dai et al., 2023; Yang et al., 2025; Alayrac et al., 2022; Su et al., 2023; Wu et al., 2024), the mechanisms by which these models process and synchronize non-textual data across heterogeneous sampling rates remain poorly understood. While specialized models excel at generating individual modalities such as video (Kondratyuk et al., 2024; Yan et al., 2021) or speech (Borsos et al., 2023), unified LLMs may still struggle to map their linguistic strengths to fine-grained temporal dependencies in visual and acoustic domains (Sakshi et al., 2024; Kondratyuk et al., 2024). Deeper investigation is needed, not only into the performance of these architectures, but also into the factors that impact decoder-only multimodal synchronization performance across inputs and outputs.

Understanding these mechanisms requires a relatively controlled setting that reduces synchronization from confounding architectural factors. Existing large-scale systems such as Qwen2.5-Omni (Xu et al., 2025a) and AudioFlamingo (Goel et al., 2025) operate at a scale where mixture-of-experts layers, multi-stage training pipelines, and massive dataset mixtures make it impossible to attribute synchronization behavior to specific design choices. For instance, key factors such as positional encoding design, modality ordering, and handling heterogeneous frame rates can all affect synchronization quality. Whether explicit timestamp-aware schemes are necessary, or whether simpler implicit indexing mechanisms suffice, cannot be answered by studying these highly complex systems alone. A cleaner testbed is needed.

We use a minimal unified decoder, dubbed, `Visatronic` as an experimental platform deliberately avoiding mixture-of-experts routing and multi-stage training, so that specific design choices—position-ID strategy and modality ordering can be varied in isolation. We also focus on a *controlled "playground" task*: video-text to speech (VTTS) synthesis—generation of speech conditioned jointly on a video of a talking people and its corresponding text transcript, making synchronization quality directly measurable. The ordering and position-ID variants are the independent variables in our study, not deployment-oriented engineering optimizations. VTTS is well suited for this investigation for four reasons: (i) it is a representative generative task with complementary multimodal input—text disambiguates homovisemes while video provides temporal and expressive signals absent from text; (ii) it involves heterogeneous sampling rates (sparse text, video frames, continuous speech) from a synchronized source, making synchronization quality directly measurable; (iii) successful generation requires fine-grained temporal dependencies between visual articulatory cues and acoustic output; and (iv) large-scale diverse datasets enable rigorous benchmarking. Our empirical investigation of `Visatronic` on VTTS task yields four key findings.

**Decoder-only models can leverage complementary multimodal information.** Systematic ablations show that, in a unified decoder-only architecture, removing either text or video causes large degradation, confirming that the model uses both modalities rather than collapsing to a single-source shortcut. The contributions are asymmetric: text is more critical for lexical content, while video provides temporal and expressive cues that improve synchronization and can improve optimization efficiency under specific ordering choices (Section 4, Table 6).

**Decoder-only models can synchronize heterogeneous sampling rates through position-ID design.** As an alternative to explicit timestamp-aware schemes such as TMRoPE (Xu et al., 2025a), we show that a unified decoder-only model can bridge heterogeneous rates using standard positional embeddings together with position-ID assignment. In our co-temporal ordered indexing setup, video IDs are mapped to the speech-time index to provide a shared temporal reference; in global sequential indexing, IDs follow concatenation order without explicit cross-modal temporal correspondence (Table 4). This shows that synchronization in decoder-only VTTS is sensitive to position-ID design without specialized alignment modules or explicit timestamp tokens in the model input. We additionally show that modality ordering further affects synchronization, with the optimal ordering depending on both evaluation metric and target domain: text-first ordering generalizes more robustly across domains, while video-first ordering is stronger in-domain, indicating a trade-off between domain specialization and transferability (Section 4, Tables 3 and 4).

**Synchronization mechanisms transfer to out-of-domain data.** `Visatronic`, trained exclusively on VoxCeleb2, generalizes zero-shot to LRS3–outperforming directly comparable models trained on LRS3 without domain-specific fine-tuning, lip detection, or architectural adaptation. We attribute this to the model learning transferable cross-modal structure rather than only dataset-specific shortcuts. Diverse training conditions (in-the-wild, noisy) may contribute to this transfer behavior. Furthermore, modality ordering affects generalization: text-first ordering transfers more robustly to unseen videos than video-first ordering, consistent with lexical content being less sensitive to visual domain shift than appearance statistics (Section 4.2, Table 3).

**Evaluating decoder-only synchronization requires fine-grained metrics.** To analyze synchronization behavior in a unified decoder-only model, we introduce `TimeSync`, a phoneme-level metric based on forced alignment that measures absolute temporal offsets between corresponding phonemes in generated and reference speech. This enables diagnosis of *where* and *by how much* synchronization breaks down (e.g., "the /s/ phoneme is 120ms early"). In contrast, widely used frame-level metrics such as LSE-D/LSE-C (Prajwal et al., 2020b) and LMD (Chen et al., 2019) produce aggregate scores and cannot localize errors to specific phonemes. Thus, `TimeSync` complements metrics such as WER by making decoder-only temporal failure modes directly interpretable (Section 3, Section 4 Figure 8 and Table 4).

We hope our findings provide actionable insights for designing unified decoder-only architectures for multimodal tasks requiring fine-grained temporal reasoning across heterogeneous modalities.

## 2 Visatronic

### 2.1 Video-Text To Speech (VTTS)

Video-text-to-speech synthesis is formulated as follows. Given (a) input video frames of a speaker $\mathbf{x}^v \in \mathbb{R}^{T^v \times H \times W \times 3}$, where $H$ and $W$ denote spatial video resolution (frame height and width, respectively) and $T^v$ is the total number of frames in the video; and (b) text tokens $\{\mathbf{x}_i^t\}_{i=1}^N$, representing the transcript of speech in the video where $\mathbf{x}_i^t \in Vocabulary$ and $N$ is length of the tokenized transcript, the goal is to generate speech $\mathbf{x}^s \in \mathbb{R}^{T^s}$, where $T^s$ is length of speech signal, such that (i) spoken content matches the text $\{\mathbf{x}_i^t\}_1^N$, and (ii) generated speech is temporally aligned with facial dynamics in video.

This task is challenging because it requires jointly modeling heterogeneous modalities with different structures and sampling rates: sparse symbolic text, video frames, and dense acoustic trajectories. These properties make VTTS a suitable setting for testing synchronization mechanisms introduced in Section 1, especially the interaction between token ordering and position-ID design. To isolate these effects, we adopt a simple unified decoder-only modeling approach rather than specialized alignment modules.

### 2.2 Discrete Input Representation Design

**Video Representation.** To obtain a latent representation of the video input[1] $\mathbf{x}^v$, we use a pretrained VQ-VAE model (Yan et al., 2021), trained on the Kinetics-600 dataset (Carreira et al., 2018). We choose VQ-VAE over discriminative video representations (e.g., CLIP (Radford et al., 2021), AV-HuBERT (Shi et al., 2022), or VideoMAE (Tong et al., 2022)) because VTTS requires preserving fine-grained spatial and temporal facial dynamics rather than semantic or category-level alignment. Discriminative encoders are optimized for global semantic matching or classification objectives and discard spatial locality in the process, whereas VQ-VAE retains a grid of discrete visual tokens per frame that encode fine-grained cues such as mouth shape, jaw motion, and facial muscle activations–all of which are critical for modeling phoneme timing and articulatory dynamics. Furthermore, VQ-VAE's reconstruction objective encourages preservation of visual detail at the frame level, making it better suited for a generation task where fine-grained temporal correspondence between visual and acoustic signals must be learned from scratch without pretrained audio-visual supervision. Each frame $\mathbf{x}_t^v \in \mathbb{R}^{H \times W \times 3}$ is encoded to a spatially downsampled latent representation $\mathbf{y}_t^v \in \mathbb{R}^{H' \times W' \times D}$. Each spatial element in $\mathbf{y}_t^v$ is then quantized to the nearest entry in the VQ-VAE's learned codebook $\mathbf{C}^v = \{\mathbf{c}_1^v, \ldots, \mathbf{c}_{K^v}^v\}$ using $\ell_2$ distance, resulting in a discrete token grid $\mathbf{v}_t \in [\mathbb{C}^v]^{H' \times W'}$. This

---

[1]In the rest of the paper, we denote tensors as $\mathbf{x}$ while $\mathbf{x}_{i,\ldots}$ denotes the $(i,\ldots)$-th component of tensor $\mathbf{x}$.

quantization process is illustrated in Figure 2. The VQ-VAE compresses each $224 \times 224$ frame into a $16 \times 16$ grid using a codebook of size $K^v = 2048$ and embedding dimension $D = 3264$. These discrete tokens are then mapped to continuous embeddings $\mathbf{e}^v_{t,h,w} \in \mathbb{R}^{D'}$ via a learnable embedding layer $\mathbf{E}^v : \mathbb{C}^v \to \mathbb{R}^{D'}$, resulting in an embedding map $\mathbf{e}^v_t \in \mathbb{R}^{16 \times 16 \times D'}$. We explore multiple spatial aggregation functions over this grid (Section 4, Table 5) and select summation based on the ablation results: $\mathbf{z}^v_t = \sum_{h=1}^{16} \sum_{w=1}^{16} \mathbf{e}^v_{t,h,w}$. This final vector $\mathbf{z}^v_t$ serves as a compact, information-rich embedding of the frame that preserves spatial and speaker-specific features while aligning to the decoder's input dimension.

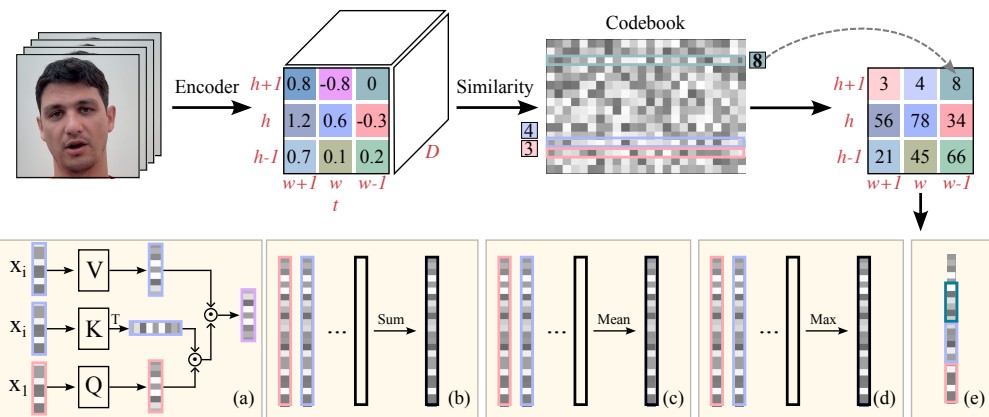

Figure 2: **Video representation.** Each video frame at time $t$ is encoded via a VQ-VAE (Yan et al., 2021) into a downsampled spatial grid in $\mathbb{R}^{H' \times W' \times D}$. Each vector at location $(h, w)$ is quantized to a discrete token using the learned codebook $\mathbf{C}^v$ via $l_2$ similarity. These discrete tokens are embedded into $\mathbb{R}^{D'}$ and aggregated across the spatial grid to produce the final frame-level embedding input to the transformer. See Section 2.2 for details.

**Text Representation.** For text processing, we employ a character-level tokenizer that maps the input text $\{\mathbf{x}^t_i\}_1^N$ to a sequence of discrete tokens $\mathbf{t}_j \in \mathbb{C}^t = \{1, 2, \ldots, K^t\}$ with $|\mathbb{C}^t| = K^t$, followed by a learnable embedding layer $\mathbf{E}^t(\cdot) : \mathbb{C}^t \to \mathbb{R}^{D'}$. Character-level tokenization reduces vocabulary size $K^t$ and improves generalization by capturing fine-grained linguistic features.

**Speaker Representation.** For multi-speaker modeling, we extract speaker representations using a pre-trained dvector model (Variani et al., 2014) that produces 512-dimensional embeddings. These speaker embeddings are projected through a learnable linear layer to match the model dimension $D'$, and are used to preserve speaker characteristics in generated speech.

**Speech Representation.** We use dMel (Bai et al., 2024), a discretization approach for speech processing; see Figure 3 for an overview. Given an input speech signal $\mathbf{x}^s$, we first compute continuous log mel-filterbanks $\mathbf{y}^s_t \in \mathbb{R}^F$ for a frame at time $t$, where $F$ is the number of log mel-filterbanks. Then, we map every log mel-filterbank $\mathbf{y}^s_{t,f} \in \mathbb{R}$ to a discrete value $\mathbf{s}_{t,f} \in \mathbb{C}^s = \{1, 2, \ldots 2^{K^s}\}$ using a codebook $\mathbf{C}^s = \{\mathbf{c}^s_1, \mathbf{c}^s_2, \ldots, \mathbf{c}^s_{2^{K^s}}\}$: $\mathbf{c}^s_i \in \mathbb{R}$ are evenly spaced values in the range $[m, M]$, where $m$ and $M$ are the minimum and maximum values of mel-filterbanks computed across the dataset. To discretize, we take the closest codebook value, i.e, $\mathbf{s}_{t,f} = \arg\min_{i \in \mathbb{C}^s} |\mathbf{y}^s_{t,f} - \mathbf{c}^s_i|$. After each speech frame is discretized, every discrete value is mapped via a learnable embedding layer $\mathbf{E}^s(\cdot) : \mathbb{C}^s \to \mathbb{R}^{d'}$ to a representation $\mathbf{e}^s_{t,f}$. The representation for the whole frame is given by $\mathbf{e}^s_t \in \mathbb{R}^{F \times d'}$, where $d'$ is the intermediate dimension. Subsequently, we stack these embeddings and project the resulting vector to a final embedding $\mathbf{z}^s_t \in \mathbb{R}^{D'}$ via a learnable linear layer $\mathbf{L}^s(\cdot) : \mathbb{R}^{Fd'} \to \mathbb{R}^{D'}$: $\mathbf{z}^s_t = \mathbf{L}^s([\mathbf{e}^s_{t,1}, \mathbf{e}^s_{t,2}, \ldots, \mathbf{e}^s_{t,F}])$. This training-free discretization enables effective processing of speech signals in our framework. Following Bai et al. (2024) we use $K^s = 4$ bits with $|\mathbb{C}^s| = 16$, $F = 80$ log mel-filterbank channels and $d' = 24$.

**Speech Inversion.** To reconstruct the speech signal $\mathbf{x}^s$ from the speech discrete values $\mathbf{s}_{t,f}$ predicted by the multimodal transformer decoder (Section 2.3), we follow Bai et al. (2024): first, we transform the indices back

Figure 3: **Speech representation.** We follow the speech discretization process from dMel by Bai et al. (2024): each continuous mel-filterbank at time $t$ extracted from the raw audio is mapped into a discrete values using a codebook of evenly spaced values. Afterwards, each discretized log mel-filterbank at time $t$ is mapped through a learnable embedding layer, all representations for log mel-filterbanks at time $t$ are stacked together and resulting vector is projected by a learnable linear layer to the model dimension $D'$. All discretized log mel-filterbanks at time $t$ are predicted in parallel and independently.

to the log mel-filterbanks via the codebook $\mathbf{C}^s$: $\hat{\mathbf{y}}_{t,f}^s = \mathbf{c}_{\mathbf{s}_{t,f}}^s$. Subsequently, we apply a vocoder (Yamamoto et al., 2020) to transform reconstructed log mel-filterbanks $\hat{\mathbf{y}}_{t,f}^s$ back into the time domain signal $\mathbf{x}^s$. The vocoder is trained independently and is not part of the `Visatronic` model.

## 2.3 Unified Multimodal Video-Text-Speech Transformer Decoder

We use a unified multimodal decoder-only transformer architecture for processing multiple modalities – video, text and speech – in order to generate speech given video and text inputs, see Figure 1. The architecture consists of a single transformer decoder that processes **discrete** multimodal input representations from Section 2.2. Unlike traditional approaches that use one modality as input, or separate encoder(s) for multimodal input, our unified architecture enables cross-modal interactions through self-attention layers while maintaining temporal coherence.

While we leverage general-purpose pretrained models for modality-specific preprocessing (VQ-VAE for video encoding (Yan et al., 2021), dvector for speaker embedding (Variani et al., 2014), and vocoder for speech reconstruction (Yamamoto et al., 2020)), these components remain frozen and serve purely as feature extractors. Critically, our approach does not require task-specific audio-visual encoders (e.g., AV-HuBERT (Shi et al., 2022)), emotion recognition networks, or explicit fusion modules as in recent video-to-speech methods (Choi et al., 2025). Instead, all trainable parameters reside in a single unified decoder that jointly learns cross-modal alignment and speech generation through self-attention, without additional task-specific alignment modules.

The model is trained end-to-end using cross-entropy loss to predict in parallel all channels of the next discrete token in sequence, allowing it to learn intrinsic relationships across modalities that are crucial for tasks requiring multimodal understanding. During inference, the model can generate tokens autoregressively while maintaining coherence across all modalities.

**Integration of Multimodal Sequences.** For effective processing of multiple modalities with different temporal resolutions, we implement various input mixing strategies, see Figure 4. Systematically exploring these strategies allows us to understand how token ordering and temporal alignment affect cross-modal learning in unified decoders. The fundamental challenge lies in handling different sampling rates and temporal ordering: speech inputs from dMel are sampled at 25ms intervals $(0.00s, 0.025s, 0.05s, \dots )$, whereas 25fps video inputs are sampled at 40ms intervals $(0.00s, 0.04s, 0.08s, \dots )$, and text tokens appear sparsely in the sequence. We explore three strategies for combining multimodal sequences, differing in both token ordering and position ID assignment. These two dimensions–ordering and position indexing–are orthogonal design choices that we study independently (see Figures 4and 6).

- **Prefix strategies with timestamp-informed IDs (TV-CoTemporal, VT-Scaled).** These variants use full-prefix conditioning (all non-speech tokens appear before speech) but differ in ordering and position-ID semantics. **TV-CoTemporal** (text → video → speech) uses true co-temporal overlap: video and speech tokens that correspond in time are assigned the same position IDs, creating a shared temporal axis without explicit timestamp tokens. **VT-Scaled** (video → text → speech) uses timestamp-scaled video IDs; text IDs occupy contiguous positions inside the conditioning prefix (e.g., TV order: text $[1, \dots , L_t]$; VT order: text

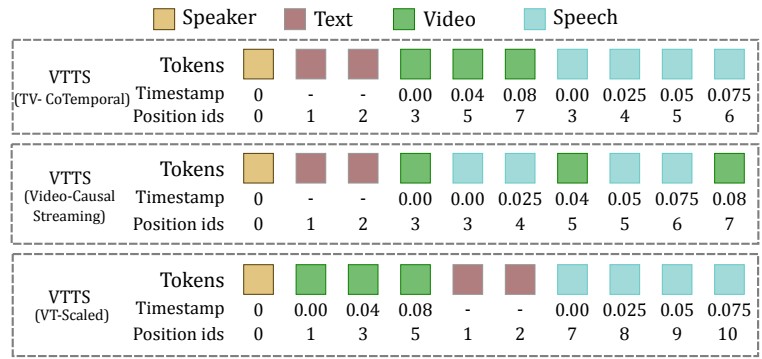

Figure 4: **Input sequence for `Visatronic`.** We encode all modalities into a discrete token space (see Figures 2 and 3), which is directly consumed by the decoder-only transformer. Each modality's discrete representation is indicated by a colored square. Each row illustrates a different token-order strategy for combining multimodal information: (top) TV-CoTemporal, where text precedes video, then speech, with true video-speech position-ID overlap for co-temporal tokens; (middle) Video-Causal-Streaming, where text appears first while speech and video are interleaved in temporal order such that speech generation at time $t$ attends to the full text and only past video tokens at $t' < t$; (bottom) VT-Scaled, where video precedes text, then speech, with timestamp-scaled video IDs but no direct video-speech ID overlap. Global sequential counterparts (TV-Global, VT-Global) are shown in Figure 6.

$[L_v + 1, \ldots, L_v + L_t]$), and speech IDs start sequentially after the full prefix. Therefore, video and speech do not share IDs in this variant. The ordering defines the autoregressive history available at each decoding step and influences cross-modal dependency learning with implications for both in-domain performance and cross-domain generalization (Section 4).

- **Global Sequential Indexing (TV-Global, VT-Global).** For each prefix ordering above, we evaluate a global-indexing ablation. Here, all tokens receive strictly increasing position IDs by concatenation order. For TV-Global: speaker $[0]$, text $[1, \ldots, L_t]$, video $[L_t+1, \ldots, L_t+L_v]$, speech $[L_t+L_v+1, \ldots, L_t+L_v+L_s]$. For VT-Global: speaker $[0]$, video $[1, \ldots, L_v]$, text $[L_v+1, \ldots, L_v+L_t]$, speech $[L_v+L_t+1, \ldots, L_v+L_t+L_s]$. This scheme does not encode cross-modal temporal correspondence in position IDs, allowing direct comparison against TV-CoTemporal and VT-Scaled.

- **Streaming (Video-Causal-Streaming).** Text tokens appear first, followed by video and speech tokens interleaved in their original temporal order, such that the speech token at time $t$ attends only to past video tokens at $t' < t$. This enforces strict causal constraints between video and speech–the model cannot attend to future video frames when generating speech–reducing attention overhead while preserving temporal progression of both modalities. Position IDs follow the temporal order, reflecting the natural arrival order of tokens in a streaming setting.

**Positional Encoding.** With sequences longer than typical TTS and heterogeneous sampling rates across modalities, positional encoding design becomes critical. Prior work has consistently shown that relative positional embeddings outperform absolute embeddings for long sequences (Touvron et al., 2023; Bai et al., 2024). We apply RoPE (Su et al., 2024) across the entire sequence for its extrapolation properties and efficiency. The three strategies above differ specifically in how position IDs are assigned across modalities, allowing us to isolate the role of position IDs in encoding cross-modal temporal alignment–independent of token ordering effects.

**Initialization.** Placing all modalities' inputs into one sequence for the decoder, we found that having different submodules to map each modality to the shared space leads to inconsistency of the embeddings across modalities (e.g. they have very different norm magnitudes). Thus, proper initialization of these submodules is essential. We identified that a proper scale for the initial weights distribution by bringing all inputs' final embeddings to the same sphere is sufficient for stable and fast training convergence.

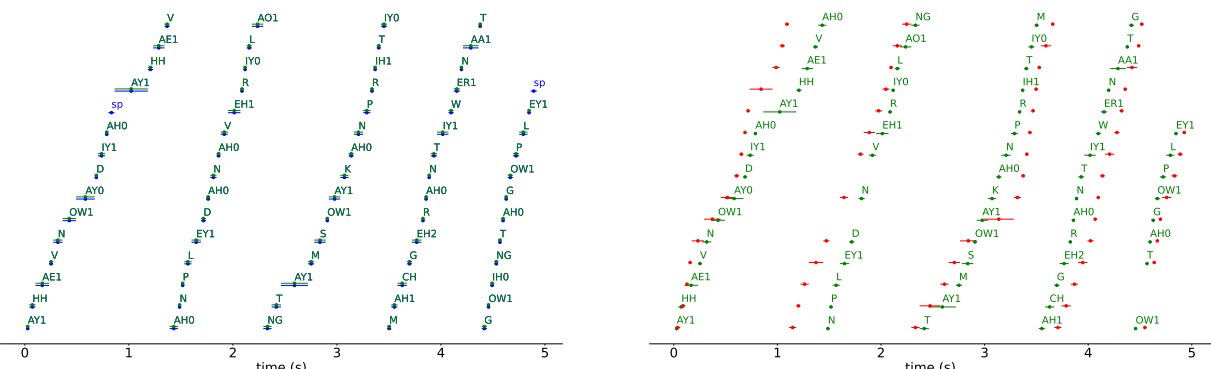

Figure 5: **TimeSync.** Visualization of phoneme-level alignment used for computing the `TimeSync`. Left: alignment in ground truth audio before (blue) and after (green) removing silence ("sp") segments. Right: aligned phoneme positions between ground truth (green) and generated (red) audio, where `TimeSync` is computed as the absolute difference between segment centers (measured in seconds).

**Model Training.**  Our unified decoder model is trained to predict speech discrete representations while being conditioned on all modalities during inference. During training we compute the cross-entropy loss $\mathcal{L}_{CE}$ only on the speech discrete representations, omitting the loss on others. All $F$ discrete log mel-filterbanks at each timestamp $t$ are predicted independently and in parallel. To ensure robust training, we follow dMel training observations and apply random span masking with probability $p$ to video, text and speech representations, forcing the model to leverage cross-modal information rather than relying solely on one modality. Speech masked regions are excluded from the loss computation. During inference, the model autoregressively generates speech discrete representations while being conditioned on speaker information, video and text.

> **Design Highlights:**  (i) single unified decoder that jointly processes speaker, video, text, and speech discrete tokens without task-specific fusion modules; (ii) controlled sequence-design decomposition into token ordering (TV-CoTemporal, VT-Scaled, Video-Causal-Streaming) and position-ID assignment (co-temporal overlap, timestamp-scaled prefix IDs, global sequential IDs), enabling clear ablations of synchronization mechanisms; (iii) robust training with random span masking and speech-only cross-entropy supervision, which discourages single-modality shortcuts and is intended to promote cross-modal dependency learning; (iv) embedding-scale initialization that aligns modality projection magnitudes and is used to improve optimization stability and convergence speed.

## 3   TimeSync

To evaluate how well the generated speech aligns temporally with the ground truth, we define a phoneme-level alignment metric, `TimeSync`. It is computed as:

$$\texttt{TimeSync} = \frac{1}{N} \sum_{\phi^{GT}} \left| t^{model}_{f(\phi^{GT})} - t^{GT}_{\phi^{GT}} \right| \tag{1}$$

where $N$ is the total number of phonemes $\phi^{GT}$ in the ground truth transcriptions obtained from the original audio samples. $t^{GT}_{\phi^{GT}} = (start_{\phi^{GT}} + end_{\phi^{GT}})/2$ denotes the segment's center (in seconds) for each ground truth phoneme. $f(\phi^{GT})$ denotes the corresponding aligned phoneme in the generated audio, and $t^{model}_{f(\phi^{GT})} = (start_{f(\phi^{GT})} + end_{f(\phi^{GT})})/2$ is the center of that segment. Here, $start$ and $end$ indicate the phoneme's start and end timestamps in either the generated or ground truth audio. This metric captures the temporal deviation between aligned phoneme centers and helps quantify how well the model preserves synchronization.

In practice, `TimeSync` is computed by taking ground truth transcription and doing force alignment of its phoneme sequence to audio (either generated or original audio) via forced alignment using an HMM model (we use https://github.com/richardbaihe/a3t from Bai et al. (2022)); we use the HTK toolkit (Young et al., 2002) as our implementation, though the metric itself is toolkit-agnostic. This procedure gives us phoneme location in time and phoneme duration for each audio. Afterward, we exclude silence ("sp") and its duration from each alignment (Figure 5, left), as our current metric focuses on spoken-phoneme timing rather than pause timing. Because every word has several possible phoneme sequences, we use Levenshtein distance to align phoneme sequences obtained for generated and original audio: we consider phonemes aligned if they are equal or related by substitution. Then, we compute the average absolute time difference between centers of aligned phoneme segments in generated and original audio (Figure 5, right). Finally, we compute the average absolute time difference between locations of centers of the phoneme segments for ground truth and generated audio and average across all phonemes in the test set.

> **`TimeSync` Enables:** (i) phoneme-level diagnosis of where synchronization breaks down; (ii) absolute temporal offset measurement without warping artifacts; (iii) fine-grained assessment of temporally salient misalignments, whereas frame-level aggregate metrics (e.g., LSE-D/LSE-C) do not provide phoneme-level localization.

## 4 Experiments

We systematically evaluate `Visatronic` across four axes: modality contributions, positional encoding and ordering strategies, and cross-domain generalization. Our evaluation combines objective metrics (WER, `TimeSync`), subjective human assessment (MOS), and ablation studies to understand not just *what* the model achieves but *how* its architectural choices drive performance. Comparisons against prior work serve to confirm that cross-modal information flow works as intended, rather than to claim superiority over specialized systems. We train on VoxCeleb2 and evaluate zero-shot transfer to LRS3.

**Datasets.** 1) **LRS3** (Afouras et al., 2018) is audio-visual dataset in English, compiled from TED and TEDx video presentations. 2) **VoxCeleb2** (Chung et al., 2018) is a large-scale audio-visual dataset primarily designed for speaker recognition task but applicable to various audio-visual processing domains. Firstly, we develop a pipeline for pseudo-labeling (PL) the speech using Demucs (Defossez et al., 2020) for speech enhancement, Whisper-large v2 (Radford et al., 2023) for automatic transcription, and proper data filtering as data are multilingual and without any text annotation. The initial version of labeled data, PL.v1, was obtained by keeping English-only detected samples. Later, we improved upon it by additional filtering of inconsistent too long or too short transcriptions, resulting in a cleaner PL.v2 version.

**Objective Evaluation Metrics.** To evaluate generated speech quality, we use two objective metrics. **Word Error Rate (WER)** measures intelligibility by comparing Whisper-large v2 transcriptions of generated speech against ground truth transcripts, capturing how well the model preserves linguistic content. `TimeSync` (Section 3) measures phoneme-level temporal offsets through forced alignment of phoneme sequences to audio using an HMM model from HTK (Young et al., 2002), providing finer-grained synchronization diagnosis than established frame-level metrics such as LSE-D and LSE-C (Prajwal et al., 2020b), which produce only aggregate scores and cannot identify which specific phonemes are misaligned or by how much. Together, WER and `TimeSync` capture complementary aspects of generation quality: intelligibility and temporal synchronization respectively. We use this framing as a diagnostic interpretation rather than claiming calibrated perceptual correlation against frame-level metrics in this work.

**Subjective Evaluation Metrics.** We randomly selected 50 samples each from VoxCeleb2 and LRS3 test data for human evaluation to assess the naturalness, intelligibility, synchronization and emotional expressiveness of the generated speech following Yemini et al. (2024). Using mean opinion score (MOS) with 95% confidence intervals, human evaluators rated the speech naturalness, intelligibility, synchronization and emotional expressiveness on a scale of 1 to 5, where 1 represents the worst and 5 the best quality. Details on the full protocol as well as implementation and training details are provided in Appendix.

Table 1: **Human evaluation on LRS3.** Mean opinion scores (MOS) (1-5) with 95% confidence intervals for our VoxCeleb2-trained models evaluated on LRS3. VTTS (TV-CoTemporal) achieves better performance in Intelligibility (3.62) and Synchronization (3.12), while both methods perform equally well in Naturalness (3.01). GT is an upper bound, while GT (discrete) shows degradation due to speech quantization and vocoding artifacts. These results demonstrate our model maintains good perceptual quality even on out-of-distribution data. For baseline methods, we use publicly available model checkpoints from their respective repositories to generate speech samples, which we then evaluate using our human evaluation protocol. "-" indicates unavailable metrics under the same standardized setup; this applies to dubbing baselines where only naturalness scores were available.

| Method | Intelligibility (↑) | Naturalness (↑) | Synchronization (↑) |
|---|---|---|---|
| GT | 4.79 ±0.05 | 4.79 ±0.05 | 4.73 ±0.06 |
| GT (discrete) | 4.32 ±0.11 | 3.80 ±0.11 | 4.59 ±0.07 |
| *Lip to Speech Synthesis* | | | |
| Lip2Speech (Kim et al., 2023) | 2.21 ±0.08 | 2.20 ±0.09 | 2.69 ±0.08 |
| SVTS (Mira et al., 2022) | 2.17 ±0.08 | 2.15 ±0.09 | 2.71 ±0.09 |
| VCA-GAN (Kim et al., 2021) | 2.19 ±0.08 | 2.20 ±0.09 | 2.71 ±0.08 |
| *Dubbing Methods* | | | |
| HPMDubbing (Cong et al., 2023) | - | 2.37 ±0.14 | - |
| StyleDubber (Cong et al., 2024) | - | 1.79 ±0.12 | - |
| *VTTS* | | | |
| TV-CoTemporal | **3.62 ±0.20** | **3.01 ±0.22** | **3.12 ±0.27** |
| VT-Scaled | 3.30 ±0.21 | **3.01 ±0.17** | 2.35 ±0.22 |

Table 2: **Human evaluation of emotional expressiveness on VoxCeleb2.** Mean Opinion Scores (1-5) with 95% confidence intervals for emotional consistency between video and speech (*video-speech emotions*), perceived emotion quality in speech alone (*speech emotions*), and specific emotion categories (*Angry, Happy, Fearful*). VTTS (TV-CoTemporal) achieves the highest score in video-speech emotion alignment (3.79), while VTTS (VT-Scaled) performs best in overall speech emotion quality (3.39) and in all individual emotion categories. These results demonstrate the advantage of visual conditioning for expressive speech generation. Ground truth (GT) serves as the reference, and GT (discrete) provides the upper bound achievable with our quantization-based approach. Per-category labels were collected only for GT, TTS, and VTTS (VT-Scaled) due annotation-budget constraints.

| Method | video-speech emotions (↑) | speech emotions (↑) | Angry (↑) | Happy (↑) | Fearful (↑) |
|---|---|---|---|---|---|
| GT | 4.62 ±0.07 | 4.92 ±0.04 | 4.36 | 4.43 | 4.20 |
| GT (discrete) | 4.41 ±0.10 | 4.37 ±0.12 | – | – | – |
| TTS (dmel Bai et al. (2024)) | 3.57 ±0.14 | 3.20 ±0.15 | 3.46 | 3.58 | 3.17 |
| *VTTS* | | | | | |
| Video-Causal-Streaming | 3.66 ±0.16 | 3.36 ±0.15 | – | – | – |
| TV-CoTemporal | **3.79 ±0.15** | 3.31 ±0.17 | – | – | – |
| VT-Scaled | 3.74 ±0.12 | **3.39 ±0.15** | **3.82** | **3.76** | **3.73** |

## 4.1 Human Evaluation Results

Human evaluation is presented in Table 1, where we compare our VTTS models against baselines from both lip-to-speech synthesis (Lip2Speech (Kim et al., 2023), SVTS (Mira et al., 2022), VCA-GAN (Kim et al., 2021)) and dubbing literature (HPMDubbing (Cong et al., 2023), StyleDubber (Cong et al., 2024)). Participants rated generated speech based on intelligibility, naturalness, and synchronization using 5-point mean opinion scores (MOS) with 95% confidence intervals. Entries marked "-" indicate metrics that were unavailable for those systems under a standardized protocol. Our VTTS (TV-CoTemporal) model achieves the highest scores among all: 3.62 in intelligibility and 3.12 in synchronization, and ties with VTTS (VT-Scaled) on naturalness (3.01). The synchronization gap between TV-CoTemporal (3.12) and VT-Scaled (2.35) is consistent with our ordering analysis: text-first conditioning yields stronger cross-domain synchronization robustness in human perception. These results are notably closer to the ground truth (GT) and GT (discrete) upper bounds, highlighting both the expressiveness and temporal coherence of our approach.

Table 3: **Cross-dataset generalization and validation against prior work on full-set LRS3.** Primary comparisons are reported on the full LRS3 test set for fairness. VoiceCraft-Dub* is shown for reference only and is not directly comparable because it adapts a large pretrained TTS backbone with additional AV-specific fine-tuning.

| Method | Training Dataset | WER (%) ↓ |
|---|---|---|
| V2SFlow (Choi et al., 2025) | LRS3 | 28.5 |
| LipVoicer (Yemini et al., 2024) | LRS3 | 21.4 |
| Lip2Speech (Kim et al., 2023) | LRS3 | 57.4 |
| SVTS (Mira et al., 2022) | LRS3 | 82.4 |
| VCA-GAN (Kim et al., 2021) | LRS3 | 90.6 |
| DiffV2S (Choi et al., 2023) | LRS3 | 39.2 |
| VoiceCraft-Dub* (pretrained) (Sung-Bin et al., 2025) | LRS3 fine-tuned | 1.38 |
| TTS | VoxCeleb2 | 27.8 |
| *VTTS* | | |
| VT-Scaled | VoxCeleb2 | 59.6 |
| TV-Global | VoxCeleb2 | 45.1 |
| VT-Global | VoxCeleb2 | 38.5 |
| TV-CoTemporal | VoxCeleb2 | **17.9** |
| VT-Scaled | LRS3 | 28.9 |
| TV-CoTemporal | LRS3 | **17.7** |

**Emotional Expressiveness Evaluation.**  Beyond intelligibility, naturalness, and synchronization, we evaluate whether video conditioning enables emotionally expressive speech that aligns with facial expressions. We select a subset of 50 samples with strong non-neutral emotions (using emotion2vec (Ma et al., 2024) model with >99% confidence across 6 emotion classes) and conduct additional human evaluation. Raters assess two dimensions: (1) *video-speech emotions:* how well the emotional tone in speech matches the facial expressions in the video, and (2) *speech emotions:* the perceived emotional quality and expressiveness of the speech itself. Table 2 presents results across these dimensions and specific emotion categories (Angry, Happy, Fearful) for *speech emotions*. VTTS (TV-CoTemporal) achieves the highest video-speech emotion alignment score (3.79/5.0), notably outperforming TTS (3.57/5.0) which lacks visual context. For overall speech emotion quality, VTTS (VT-Scaled) achieves 3.39/5.0 compared to TTS's 3.20/5.0, while VTTS (Video-Causal-Streaming) remains competitive (3.36/5.0), indicating that streaming constraints retain substantial emotional information. Notably, VTTS (VT-Scaled) demonstrates strong performance across all individually scored emotion categories (Angry: 3.82, Happy: 3.76, Fearful: 3.73), approaching the ground truth upper bound. This split between variants is consistent with our other measured trends: TV-CoTemporal is strongest on synchronization/alignment-focused metrics (LRS3 sync MOS in Table 1 and video-speech emotion alignment in Table 2), whereas VT-Scaled is strongest on in-domain speech quality/content metrics (VoxCeleb2 WER in Table 4 and intelligibility/naturalness MOS in Table 8). We report per-category ratings only for GT, TTS, and the strongest VTTS variant (VT-Scaled) to keep annotation load tractable. These results demonstrate that visual conditioning not only improves temporal synchronization but also enables the model to generate emotionally expressive speech that naturally aligns with speakers' facial expressions, a critical capability for applications like dubbing and character animation where emotional consistency is essential.

## 4.2 State-of-the-Art Comparison

To validate that our architectural choices properly leverage multimodal information, we compare against prior work. These comparisons serve to confirm that cross-modal information flow works as intended, rather than to claim superiority over specialized systems.

**Text information utilization.** Video-only methods like V2SFlow (28.5% WER on LRS3) face inherent linguistic ambiguity from homovisemes. Our substantially lower error rates (17.9% WER for VTTS TV-CoTemporal on the full LRS3 set, Table 3) confirm that the unified decoder effectively uses text to resolve visual ambiguities—validating that the architectural design properly integrates linguistic information.

**Generalization from diverse training.** On full-set LRS3, the directly comparable LRS3-trained VTTS (TV-CoTemporal) model achieves 17.7% WER, while our VoxCeleb2-trained TV-CoTemporal model reaches

Table 4: **WER and `TimeSync` on VoxCeleb2.** We report three WERs: (i) GT WER from original audio, (ii) GT (discrete) WER from reconstructed audio using GT speech tokens (lower bound), and (iii) WER from generated audio by a model. All values are computed using Whisper-large v2. The first two rows correspond to PL.v1 transcripts; the remaining rows use PL.v2. `TimeSync` is reported for the PL.v2 section. Among non-global prefix variants, VTTS (VT-Scaled) achieves 12.2% WER, while global-indexing ablations reach 12.1–12.7% WER. `TimeSync` further shows improved phoneme-level synchronization with video-conditioned models. Rows labeled "global" compare Global Sequential Indexing as an ablation against their non-global counterparts (TV-CoTemporal and VT-Scaled).

| Method | Input Modality | GT WER (↓) | GT (discrete) WER (↓) | WER (↓) | TimeSync (s) (↓) |
|---|---|---|---|---|---|
| TTS (dmel Bai et al. (2024)) | Text | | | $19.0_{+8.5}$ | - |
| *VTTS* | | $4.0 \pm 0.1$ | $10.5 \pm 0.1$ | | |
| VT-Scaled | Video-Text | | | $\mathbf{17.2}_{+6.7}$ | - |
| TTS | Text | | | $14.7_{+4.6}$ | $0.62 \pm 0.98$ |
| *VTTS* | | $2.6 \pm 0.1$ | $10.1 \pm 0.2$ | | |
| Video-Causal-Streaming | Text-Video | | | $14.5_{+4.4}$ | $0.49 \pm 0.63$ |
| LTTS (LT-Scaled) | Lip-Text | | | $14.0_{+3.9}$ | $0.46 \pm 0.61$ |
| *VTTS (Global IDs)* | | | | | |
| TV-Global | Text-Video | $2.6 \pm 0.1$ | $10.1 \pm 0.2$ | $12.7_{+2.6}$ | $0.48 \pm 0.69$ |
| VT-Global | Video-Text | | | $\mathbf{12.1}_{+2.0}$ | $0.46 \pm 0.61$ |
| *VTTS (Non-global IDs)* | | | | | |
| TV-CoTemporal | Text-Video | $2.6 \pm 0.1$ | $10.1 \pm 0.2$ | $14.1_{+4.0}$ | $\mathbf{0.44 \pm 0.65}$ |
| VT-Scaled | Video-Text | | | $12.2_{+2.1}$ | $0.47 \pm 0.63$ |

17.9% WER zero-shot (Table 3). This near-parity without any LRS3 training indicates that diverse large-scale pretraining can transfer effectively across domains.

In Table 3, we further evaluate models trained only on VoxCeleb2. On full-set LRS3, the text-only TTS baseline gives 27.8% WER, while adding video with TV-CoTemporal reduces this to 17.9% WER. Crucially, the same training data does not guarantee gains for other synchronization schemes: TV-Global drops to 45.1%, VT-Global to 38.5%, and VT-Scaled to 59.6%. Thus, video conditioning helps only when ordering and position-ID design are compatible with cross-domain synchronization. The LRS3-trained rows follow the same direction (TV-CoTemporal[†]: 17.7% vs VT-Scaled[†]: 28.9%), reinforcing that TV-CoTemporal is the robust configuration in this comparison. For transfer-focused analysis with synchronization scores, we additionally report the introduced 3–45s LRS3 setting in Appendix Table 9, where TV-CoTemporal reaches 4.8% WER and 0.22s `TimeSync`. The relative ordering remains consistent with the full-set trends: among VoxCeleb2-trained models, TV-CoTemporal is strongest, followed by TTS and VT-Global, while VT-Scaled and TV-Global remain weaker; the LRS3-trained rows are 5.6% (TV-CoTemporal) and 6.3% (VT-Scaled). This ranking is reliable because it repeats across both evaluation protocols reported in this paper (full-set LRS3 and the 3–45s transfer-focused setting) and also matches the ordering within the LRS3-trained pair. VoiceCraft-Dub* (1.38% WER) is shown for context only and is not directly comparable: it builds on a large pretrained VoiceCraft TTS model, uses Encodec-style discrete speech representations, and applies additional audio-visual fusion/adaptation during fine-tuning, whereas our VTTS models are trained from scratch under a unified decoder setup.

Table 4 shows a comparison of our proposed models and the TTS baseline trained and evaluated on VoxCeleb2. The first two rows are PL.v1 results kept for continuity with earlier versions of the dataset; all remaining rows use PL.v2 and form the main comparison set (including all reported `TimeSync` values). All results demonstrate that incorporating video improves both speech content generation and time synchronization. For `TimeSync` computation, PL.v2 is treated as a ground truth for VoxCeleb2. We evaluate performance using three types of WER. GT WER is computed on original audio using Whisper and serves as an ASR baseline. GT (discrete) WER is computed on audio reconstructed from ground truth discrete speech tokens using our vocoder, representing the lower bound set by quantization and vocoding artifacts. WER is the main evaluation target, measuring how well the model-generated speech aligns with the ground truth transcript.

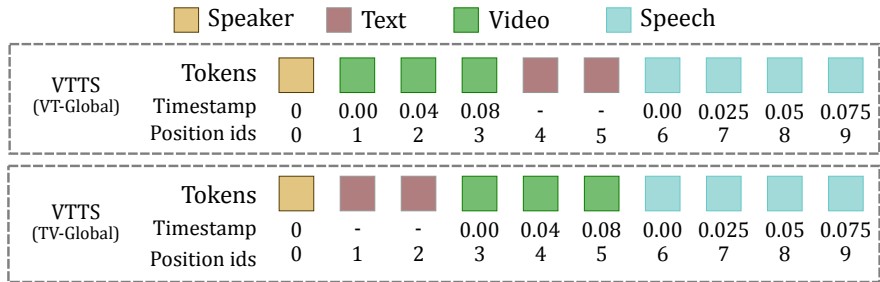

Figure 6: **Ablation over sequence-design variants.** Visualization of performance trends across the ordering and position-ID variants analyzed in this section. Exact metric values are reported in Tables 4 and 3.

Our objective is to minimize the gap between WER and GT (discrete) WER, as this gap reflects modeling limitations beyond the quantization-vocoder pipeline. Improved annotations in PL.v2 lead to consistently better results compared to PL.v1. Furthermore, we compare our full-video VTTS model to a variant using lip crops (LTTS). The VT-Scaled VTTS variant achieves 12.2% WER, while corresponding LT-Scaled LTTS gives 14.0% WER, and the TTS baseline yields 14.7% WER. These results show that using full-face video provides richer information and that our model effectively leverages it without requiring lip cropping.

Beyond word accuracy, temporal alignment is critical for video-conditioned speech generation. As shown in Table 4, video-conditioned models consistently achieve lower `TimeSync` error than the text-only TTS baseline. Specifically, VTTS (VT-Scaled) reduces the average phoneme-level timing error from $0.62 \pm 0.98$ s (TTS) to $0.47 \pm 0.63$ s, indicating substantially tighter synchronization with the visual stream. Importantly, this improvement is not limited to mean performance: the reduced variance demonstrates that VTTS produces more stable alignment across diverse utterances, whereas TTS exhibits occasional large timing deviations that are perceptually disruptive when overlaid on video. Comparing different conditioning strategies, both VTTS (Video-Causal-Streaming) and VTTS (TV-CoTemporal) outperform TTS, confirming that access to visual information provides explicit temporal cues unavailable to text-only models. Furthermore, the comparable `TimeSync` scores between full-face VTTS (0.44–0.47 s) and lip-only LTTS (0.46 s) suggest that lip motion provides the primary temporal signal, while full-face context offers a modest but consistent additional benefit. Overall, these results validate `TimeSync` as a sensitive metric for evaluating audio-visual synchronization and demonstrate that explicit video conditioning leads to more precise phoneme-level alignment than text-only synthesis.

### 4.3 Ablations

**Effect of Token Ordering and Position ID Assignment.** Figure 6 summarizes the ablation results for the ordered strategies introduced in Section 2.3. Across all evaluations, a consistent pattern emerges among non-global prefix variants: VTTS (VT-Scaled) achieves stronger in-domain performance (12.2% WER on VoxCeleb2, Table 4) and faster convergence (Table 7), while TV-CoTemporal generalizes more robustly to unseen domains (17.9% vs 59.6% WER on full-set LRS3 zero-shot, Table 3), consistent across both objective and subjective metrics (Table 1). Position ID assignment interacts asymmetrically with ordering: under VT ordering, *timestamp-scaled prefix IDs* (VT-Scaled) marginally underperform *global sequential indexing* (VT-Global) on both WER and `TimeSync` (12.2% vs 12.1%, 0.47s vs 0.46s), whereas under TV ordering, *global sequential indexing* (TV-Global) improves WER (12.7% vs 14.1%) while *co-temporal overlap* (TV-CoTemporal) improves `TimeSync` (0.44s vs 0.48s). These margins are small in some cases, but they consistently indicate that ordering and position ID assignment are not independent design choices. TV-CoTemporal performs better under transfer because it reduces cross-modal temporal distance, enabling stronger early-layer alignment. This happens because co-temporal position-ID overlap places video and speech tokens on a shared temporal axis, so alignment cues are matched at the same positional index instead of being separated by long prefix offsets. In contrast, VT-Scaled and VT-Global perform better in-domain because video-first prefixing and sequential IDs favor stable local fitting to in-domain lexical-acoustic statistics. This suggests a practical recommendation: use TV-CoTemporal for generalization related applications such as dubbing unseen speakers, and VT-Scaled for in-domain applications.

Table 5: **Effect of video aggregation strategies.** Summation performs best, with a small margin over max pooling (12.2% vs 12.4%).

|  | Attention | Max | Stacking | Sum |
|---|---|---|---|---|
| WER | 14.5 | 12.4 | 14.3 | **12.2** |

Table 6: **Modality ablations.** Both modalities are necessary because removing either modality causes large WER increases: VT-Scaled rises from 12.2 to 74.5 (w/o T) or 46.4 (w/o V), and TV-CoTemporal rises from 14.1 to 112.0 (w/o T) or 100.7 (w/o V).

| Method | GT WER (↓) | GT (discrete) WER (↓) | WER (↓) |
|---|---|---|---|
| *VTTS* | | | |
| VT-Scaled | | | **12.2** |
|   w/o T | 2.6 ±0.1 | 10.1 ±0.2 | 74.5 |
|   w/o V | | | 46.4 |
| TV-CoTemporal | | | 14.1 |
|   w/o T | 2.6 ±0.1 | 10.1 ±0.2 | 112.0 |
|   w/o V | | | 100.7 |

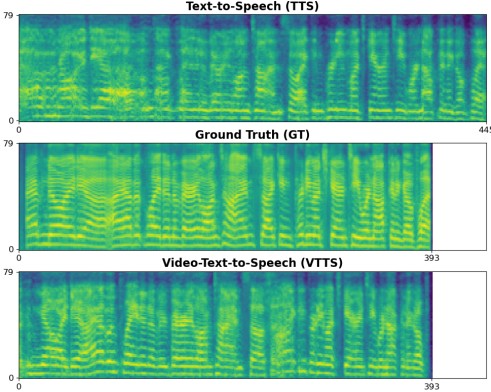

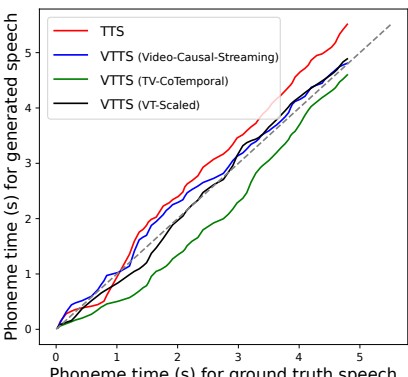

Figure 7: **Spectrogram analysis (left):** Log mel-spectrogram comparison for TTS (top), GT (middle), and VTTS (VT-Scaled, bottom). VTTS (VT-Scaled) better matches GT's timing (393 frames) and energy patterns, unlike TTS which overextends (445 frames). **Qualitative alignment (right):** `TimeSync` visualization of phoneme alignment for the same example. Ground truth and generated phoneme segment centers are plotted on $x$- and $y$-axes respectively, with ideal sync shown as a gray diagonal. VTTS models stay closer to the diagonal than TTS, reflecting better temporal alignment.

**Aggregation of Video Representations.** Table 5 shows results of different strategies for spatial aggregation of video representations before inputting into the VTTS (TV-CoTemporal) decoder. Summation achieves the best WER (12.2%), but the margin over max pooling (12.4%) is small, so we treat these two as effectively comparable and choose summation for simplicity. Both significantly outperform more complex strategies like attention (14.5%) and stacking (14.3%), suggesting that simple element-wise aggregation might be sufficient for effective video input compression.

**Qualitative Results.** Figure 7 (left) shows mel-spectrogram comparisons between TTS, GT, and VTTS (VT-Scaled). The mel-spectrogram generated by VTTS (VT-Scaled) closely resembles GT in terms of temporal structure and speech patterns, particularly in capturing natural pauses and utterance duration. While TTS generates beyond the original duration (445 frames vs GT's 393 frames) and fails to maintain proper temporal alignment, VTTS (VT-Scaled) accurately matches GT's frame length (393 frames) and successfully captures speech dynamics including pause locations. This demonstrates VTTS's ability to leverage visual information for generating temporally coherent speech that aligns with the original video timing. The spectral patterns in VTTS (VT-Scaled) also show similar energy distributions to GT, particularly in the harmonic structure during speech segments. Analysis of `TimeSync` for synchronization is shown in Figure 7 (right). The VTTS curves cluster closer to the diagonal than TTS, indicating smaller phoneme-level timing offsets across the utterance rather than only at a few isolated points. TTS exhibits larger deviations, especially in later phoneme segments, consistent with the longer-duration drift seen in the spectrogram comparison.

Table 7: **Convergence analysis.** Training-iteration comparison shows faster convergence when video modality is used in addition to text. At 2M iterations, VTTS variants are comparable to or better than TTS at the same budget, and VTTS (VT-Scaled) already reaches 12.2% WER. TTS reaches 14.7% WER at 3M iterations, while VTTS (VT-Scaled) maintains a smaller gap (+2.1%) from GT (discrete) WER, indicating more efficient optimization with multimodal conditioning.

| Method | Iterations | GT WER ($\downarrow$) | GT (discrete) WER ($\downarrow$) | WER ($\downarrow$) |
|---|---|---|---|---|
| TTS | 2M | | | $17.3_{+7.2}$ |
| **VTTS** | | | | |
| TV-CoTemporal | 2M | 2.6 $\pm$0.1 | 10.1 $\pm$0.2 | $17.0_{+6.9}$ |
| LTTS (LT-Scaled) | 2M | | | $\underline{14.9}_{+3.9}$ |
| VT-Scaled | 2M | | | $\mathbf{12.2}_{+2.1}$ |
| TTS | 3M | | | $14.7_{+4.6}$ |
| **VTTS** | | | | |
| TV-CoTemporal | 3M | 2.6 $\pm$0.1 | 10.1 $\pm$0.2 | $14.1_{+4.0}$ |
| LTTS (LT-Scaled) | 3M | | | $\underline{14.0}_{+3.9}$ |
| VT-Scaled | 3M | | | $\mathbf{12.2}_{+2.1}$ |

**Modality Drop.** Table 6 shows the impact of removing individual modalities *during evaluation.* Under VT-Scaled, removing text degrades WER from 12.2% to 74.5% and removing video from 12.2% to 46.4%, confirming that both modalities contribute complementary information. This happens because the absolute degradations are large for both removals (+62.3 for w/o T and +34.2 for w/o V), so the model cannot recover good transcription quality from a single modality. The degradation is even more severe under TV-CoTemporal: removing text yields 112.0% WER and removing video yields 100.7% WER, i.e., +97.9 and +86.6 relative to its 14.1 baseline. Under TV-CoTemporal, both removals are catastrophic, indicating strong joint dependence on text and video. The gap between 112.0% and 100.7% is small relative to the overall collapse, so we avoid interpreting it as evidence that one modality dominates the other in this setting. Relative to VT-Scaled, the larger collapses in TV-CoTemporal indicate tighter cross-modal coupling, because performance fails more sharply when either conditioning stream is removed.

**Training Efficiency.** Under VT-Scaled, video conditioning improves final performance and accelerates convergence relative to TTS (Table 7). At the same 2M-step budget, VTTS (VT-Scaled) reaches 12.2% WER while TTS is still at 17.3%, and even after 3M steps TTS only improves to 14.7%. VTTS (VT-Scaled) has effectively converged by 2M (12.2% at both 2M and 3M), whereas LTTS (LT-Scaled) continues improving from 14.9% to 14.0%, indicating slower convergence for lip-only conditioning. In contrast, TV-CoTemporal at 2M (17.0%) is close to TTS at 2M, so acceleration is ordering-dependent rather than uniform across variants.

> **VoxCeleb2 PL.v2:** 1.6k hours, 6k speakers, multilingual, pseudo-labeled via Demucs + Whisper-large-v2 with quality filtering. Enables zero-shot transfer to full-set LRS3 (17.9% WER, Table 3) and substantially outperforms directly comparable video-only baselines (V2SFlow: 28.5%, LipVoicer: 21.4%; Table 3), demonstrating that dataset scale and diversity may outweigh strict domain matching.

> **Key Findings:** (i) both modalities are necessary because removing text degrades WER from 12.2% to 74.5%, while removing video degrades WER from 12.2% to 46.4% (Table 6); (ii) video conditioning improves synchronization over the text-only TTS baseline because `TimeSync` drops from 0.62s to 0.47s (Table 4); (iii) ordering shows a consistent trade-off among non-global prefix variants because VT-Scaled is stronger in-domain on VoxCeleb2 (12.2 vs 14.1% WER), while TV-CoTemporal transfers better zero-shot on full-set LRS3 (17.9% vs 59.6% WER) (Tables 4, 3); (iv) under VT-Scaled, video conditioning accelerates convergence because at 2M steps VTTS reaches 12.2% WER while TTS is at 17.3%, and even TTS at 3M remains at 14.7% (Table 7).

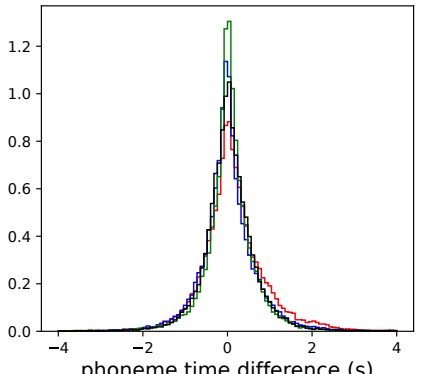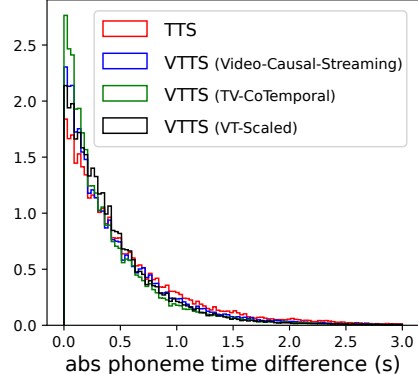

Figure 8: **Distribution of `TimeSync` phoneme timing differences.** We show the difference (left) and absolute difference (right) between ground truth and generated speech phoneme locations (center of phoneme segment) measured in seconds. The left panel shows symmetric, zero-centered distributions indicating no systematic temporal bias. The right panel reveals that VTTS variants (Video-Causal-Streaming: blue, TV-CoTemporal: green, VT-Scaled: black) concentrate phoneme timing errors within 0-0.5s, while TTS (red) exhibits heavier tails with higher probability of large timing errors (>1.0s). Video-Causal-Streaming appears intermediate between TTS and ordered variants, demonstrating superior temporal precision from video conditioning.

### 4.4 `TimeSync` Analysis

**Temporal Alignment Quality.** `TimeSync` can provide deeper insight into temporal alignment quality of the generated speech: we analyze the distribution of phoneme-level timing differences between generated and ground truth speech. Figure 8 visualizes both the signed difference (left) and absolute difference (right) between phoneme center locations in generated versus ground truth audio.

The left panel reveals that all models show approximately zero-centered distributions, indicating no systematic temporal bias (e.g., consistently generating speech too fast or too slow). However, the distribution shapes differ significantly: TTS (red) exhibits the widest spread with heavier tails, reflecting larger timing errors and inconsistent temporal alignment. In contrast, video-conditioned variants show tighter, more concentrated distributions around zero: TV-CoTemporal and VT-Scaled are the sharpest, while Video-Causal-Streaming lies in between full-ordering variants and TTS.

The right panel (absolute differences) more clearly illustrates synchronization quality. VTTS models concentrate most phonemes within 0-0.5 seconds of their ground truth timing, with probability density rapidly decreasing beyond this range. TTS shows notably higher probability mass at larger timing errors (1.0-2.0s), indicating more frequent synchronization failures. Video-Causal-Streaming again appears intermediate: better than TTS but less concentrated than TV-CoTemporal/VT-Scaled. This distribution analysis confirms that video conditioning not only improves average `TimeSync` scores (Table 4: 0.44–0.49s for VTTS variants vs 0.62s for TTS) but also reduces the variance and tail risk of synchronization errors.

Crucially, the combination of a near-zero-centered signed distribution and a heavy-tailed absolute distribution implies that failures are not primarily global rate errors; they are intermittent local breakdowns. In other words, most phonemes are well aligned, but a subset of segments exhibits large temporal slips. Combined with the per-utterance trajectories in Figure 7 (right), these large deviations are more common in later portions of an utterance, consistent with accumulated timing drift rather than uniform misalignment from the beginning.

**VTTS vs. LTTS.** We run `TimeSync` for LTTS on VoxCeleb2 test and obtain 0.46 ±0.61, whereas full-video VTTS (TV-CoTemporal) gives 0.44 ±0.65, showing a modest mean improvement from full-face conditioning over lip-only crops. Interestingly, LTTS has slightly lower variance despite worse mean; this suggests lip-only conditioning may produce more consistently biased timing, while full-face VTTS achieves better average alignment but with occasional larger deviations. Overall, the lower mean for VTTS indicates that additional facial cues beyond lips improve temporal accuracy.

## 5 Related Work

Besides *video-text-to-speech* (VTTS, our focus) setting, speech synthesis from visual input has been explored also in *video-only-to-speech* (V2S) setting. Video-only methods generate speech solely from lip movements, achieving high perceptual quality but facing inherent ambiguity from homophones: visually identical lip movements for phonetically distinct sounds (e.g., "bat" vs. "pat"). Recent V2S works trained and evaluated on LRS3 (Afouras et al., 2018), such as DiffV2S (Choi et al., 2023) (word error rate of 39.2%) and V2SFlow (Choi et al., 2025) (word error rate of 28.5%), achieve impressive naturalness but poor intelligibility, as they must resolve linguistic ambiguity from visual information alone. In applications such as movie dubbing, video game localization, audiobook production with video, or accessibility tools, ground-truth text is available alongside video. In these scenarios, *video-text-to-speech* (VTTS) can leverage both modalities to generate speech that is simultaneously intelligible (from text), temporally synchronized (from video), and expressively aligned with the speaker's facial movements. Recent VTTS work includes HPMDubbing (Cong et al., 2023) and StyleDubber (Cong et al., 2024), which use video and text as inputs but have not demonstrated cross-dataset generalization or evaluation on large-scale diverse training data like VoxCeleb2 (Chung et al., 2018). Also, no prior work has demonstrated practical zero-shot cross-dataset generalization for video-text-to-speech synthesis on large-scale diverse training data.

Speech synthesis models have made rapid progress in recent years, yet they typically focus on text-to-speech or simplified variants of video-conditioned speech generation, e.g., from cropped lip videos (Yemini et al., 2024) or generic audio generation (Kondratyuk et al., 2024). While cropped lip approaches simplify the problem through pretrained lip detectors, they omit important visual cues from the full face that contribute to accurate modeling of phoneme sounds—particularly those involving regions beyond the lips. Furthermore, many prior approaches rely on pretrained ASR (Yemini et al., 2024) or follow a pipeline of lip-reading followed by speech generation, which can degrade in noisy, multi-speaker environments where parts of the speech signal may be lost or masked by background interference. To understand how decoder-only architectures learn fine-grained temporal alignment across heterogeneous modalities, we investigate *video-text-to-speech* (VTTS) as a testbed. Unlike prior settings that use single input modalities or specialized architectures, VTTS uniquely combines video (40ms frames), text (sparse tokens), and speech (25ms frames) in a unified decoder. This enables investigation of four key architectural questions: (1) How does explicit text conditioning compare to implicit linguistic knowledge in pretrained encoders? (2) Can unified processing learn temporal alignment without task-specific modules? (3) Does training from scratch on diverse data enable better generalization than domain-specific or pretrained approaches? (4) How should synchronization be measured beyond frame-level metrics to diagnose phoneme-level timing errors? Below, we discuss how existing approaches address related problems and what questions they leave unanswered.

**Text-to-Speech (TTS).** TTS systems have evolved from early statistical approaches to end-to-end neural methods (Zen et al., 2009; Kim et al., 2020; Lee et al., 2021; Mehta et al., 2024; Popov et al., 2021; Ren et al., 2019; Shen et al., 2018). Traditional TTS faces challenges with unseen speaker styles, leading to approaches that extract speaker representations from speech data (Chen et al., 2021; Huang et al., 2022; Jia et al., 2018; Lee et al., 2022; Min et al., 2021) or incorporate face images to capture visual-acoustic correlations (Goto et al., 2020; Lee et al., 2023; Wang et al., 2022). However, static face images often neglect motion-related factors, leading to inconsistent voice generation when facial expressions vary. Recent unified architectures like VioLA (Wang et al., 2023) and VOXTLM (Maiti et al., 2024) attempt to unify speech and text modeling but rely on multi-stage hierarchical processing or lose acoustic detail through discrete content tokens. While TTS excels at generating intelligible speech from text, it lacks visual temporal dynamics, preventing investigation of cross-modal temporal synchronization and how facial expressions influence prosody.

**Lip-to-Speech (LTS).** LTS aims to reconstruct speech from silent videos of lips—crucial for scenarios with corrupted or missing audio. Early GAN-based approaches like Lip2Wav (Prajwal et al., 2020a), VCA-GAN (Kim et al., 2021), and Lip2Speech (Kim et al., 2023) demonstrated success on limited vocabulary datasets. Recent work has explored discrete token representations through AV-HuBERT (Shi et al., 2022), with ReVISE (Hsu et al., 2023) integrating HiFi-GAN for improved generation. Diffusion models have also shown promise, with LipVoicer (Yemini et al., 2024) achieving 21.4% WER on LRS3 (Afouras et al., 2018). More recently, V2SFlow (Choi et al., 2025) addresses video-only synthesis by decomposing speech

into content, pitch, and speaker attributes using rectified flow matching, achieving 28.5% WER on LRS3. While LTS methods do not take explicit text input, modern approaches implicitly encode linguistic knowledge through pretrained audio-visual encoders. For instance, LipVoicer's AV-HuBERT encoder learns to predict masked audio tokens from visual input, providing implicit phonetic structure. However, these approaches face fundamental ambiguity from *homophones*—visually identical lip movements for phonetically distinct sounds (e.g., "bat" vs. "pat"). Moreover, by focusing primarily on lip movements, they potentially overlook broader visual dynamics from the full face. Without explicit text, LTS cannot answer how explicit linguistic conditioning compares to implicit knowledge, or how models should integrate complementary modalities (text for content, video for timing) in unified architectures.

**Visual Voice Cloning and Dubbing.** Visual voice cloning (Chen et al., 2022; Jung & Kim, 2020) and dubbing (Cong et al., 2023; Hu et al., 2021; Zhang et al., 2024; Cong et al., 2024; Liu et al., 2024) address related challenges but operate under different constraints. Voice cloning typically aims to replicate a speaker's vocal identity and style using short reference audio-video pairs, often without explicit transcript supervision. Dubbing methods like HPMDubbing (Cong et al., 2023) and StyleDubber (Cong et al., 2024) replace original speech with new content in different voices or languages, relying on reference speech and specialized modules for duration, pitch, and energy prediction. Recent work has also explored VTTS through transfer learning: VoiceCraft-Dub (Sung-Bin et al., 2025) fine-tunes pretrained VoiceCraft (TTS) with audio-visual fusion layers to achieve 1.38% WER on LRS3. While this demonstrates effective adaptation of pretrained models, it requires access to large-scale pretraining infrastructure and employs complex Encodec-based representations. While these approaches achieve strong in-domain performance, their reliance on reference audio, pretrained models, or handcrafted alignment modules prevents investigation of whether unified end-to-end architectures can learn robust temporal alignment from scratch on diverse data without such scaffolding.

## 6 Limitations

Further limitations, including representation-ceiling effects from discretization and vocoding, are discussed in Appendix B.

## 7 Conclusion

We investigate a unified decoder-only transformer for video-text-to-speech that processes visual dynamics, textual content, and speech as discrete tokens in a shared embedding space. Our results show that unified multimodal optimization leverages complementary information: text is important for lexical content, while video provides temporal and expressive cues that improve synchronization, emotional alignment, and intelligibility when combined with text. We also identify a consistent ordering trade-off: video-first ordering gives stronger in-domain performance, while text-first ordering transfers more robustly across domains. Training on diverse, noisy VoxCeleb2 enables strong zero-shot generalization to full-set LRS3 (17.9% WER), outperforming prior video-only baselines trained on LRS3 (e.g., LipVoicer: 21.4%, V2SFlow: 28.5%) and reaching near-parity with the LRS3-trained TV-CoTemporal counterpart (17.7%). These results are consistent with the view that diverse large-scale training can reduce reliance on strict domain matching for robust cross-modal representations, though isolating this effect requires targeted ablations. Additionally, our `TimeSync` metric provides finer-grained phoneme-level synchronization assessment (0.47±0.63s vs. 0.62±0.98s for TTS). These findings establish VTTS as a valuable testbed for understanding how decoder-only architectures should handle heterogeneous temporal modalities.

While the approach used in our systematic analysis achieves strong performance, discretization in both speech and video can remove fine-grained cues: in practice, the speech quantization-vocoder pipeline imposes a measurable reconstruction floor, and video tokenization can suppress subtle articulatory details. The model also occasionally struggles with complex emotional expressions in challenging acoustic conditions. Future work should therefore stay centered on the decoder-only question by studying continuous (or hybrid continuous-discrete) multimodal representations within a single decoder, and by analyzing how ordering and position-ID design affect alignment when the model must process continuous temporal signals directly.

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

## Appendix

## A   Ethics Discussion

The advancement of speech technologies brings great potential but also significant ethical challenges that must not be overlooked. While we aim to create techniques that improve conditional speech synthesis for multimodal settings, it is vital to address risks proactively and promote awareness to guide responsible innovation at different levels: from researchers to the end-users. As such, we highlight several key challenges:

- **Dual-use risks**   There are always risks of impersonation, voice spoofing attacks, and fake content generation. Safeguarding and watermarking by inserting detectable markers in the generated speech are one of the quickly developing areas to detect the misuse cases.

- **Privacy**   We acknowledge the sensitivity of facial and speech data in research and technology development carrying privacy considerations, and thus, we affirm our commitment to protecting individuals' rights and fostering responsible data usage.

- **Accessibility and inclusivity**   While we are working with English-only data for the proof of concept, extending the speech technologies for diverse populations and existing spoken languages should be a top priority in the community.

- **Transparency and accountability**   Detailed documentation, limitations, analysis of failure cases, and reproducibility are essential for promoting transparency and informed usage. Responsibility in development and deployment should remain a cornerstone in the community.

## B   Limitations

While we made the best effort to tune TTS baseline, there is always a possibility we missed some details. Due to optimization issues when both modalities, video and text, are inputted into the model, we first found best hyper-parameters for our VTTS models so that the models can converge. Later, the same hyper-parameters are used for the TTS baseline by excluding video from the input into the model. However, across all experiments and hyper-parameter tuning we consistently observe that VTTS models outperform TTS models, demonstrating that video brings helpful information for the speech generation.

We did not train larger models (>300M parameters), did not use larger datasets (>1.6k hours) or pre-trained models, and leave this as a future work.

GT (discrete) WER of 10.1% defines an empirical floor for models using our current discretization pipeline, since even reconstructed ground-truth tokens incur this error. Our best VoxCeleb2 model (VTTS (VT-Scaled)) reaches 12.2% WER, only 2.1 absolute points above this floor. This indicates limited remaining headroom within the current representation pipeline and suggests that further gains increasingly depend on improving discretization and reconstruction quality, not only decoder modeling. VoxCeleb2 contains noisy and overlapping speech, which degrades mel-based discretization and vocoder reconstruction. Because the vocoder is trained independently and not optimized for in-the-wild speech, this floor remains a core limitation; future work may reduce it using neural codecs or more noise-robust vocoders.

## C   Data, Code, Reproducibility

We made the best effort to use publicly available data and official implementations (e.g. VQ-VAE for video representations). All data we used are under permissive license for research. We do our best to provide all details and steps in the main text and in Appendix. We are in the process of open-sourcing the code and releasing transcriptions PL.v2 for VoxCeleb2 data.

We do not plan to open-source any pre-trained models for sake of privacy, safety and misuse.

## D   Video Reconstruction

Although the VQ-VAE model used to extract video representations is pre-trained on the general videos, we found it reconstructs speakers videos with sufficient quality to preserve necessary spatial information. To evaluate video reconstruction quality, we employ the Fréchet Video Distance (FVD) Unterthiner et al. (2018), specifically the $FVD_{16}$ variant that assesses quality over 16-frames window. The FVD scores are computed using an I3D model trained on Kinetics-400, providing a standardized measure of video quality across different temporal scales. The FVD metric is 86.2 at resolution 64x64. *Thus, we do not finetune the model further on videos of talking people and use it as is.*

## E   Video Representation

To obtain a frame-level embedding suitable for autoregressive modeling, we aggregate the spatial grid of quantized video embeddings into a single vector $\mathbf{z}_t^v \in \mathbb{R}^{D'}$ per frame. As shown in Figure 2 (main paper), each frame is first discretized using a pretrained VQ-VAE encoder, resulting in a $H' \times W'$ grid of tokens, where each token is embedded via a learnable table $\mathbf{E}^v$. Since our unified decoder processes one token per time step, we aggregate the $H' \times W'$ spatial embeddings into a single vector before feeding into the decoder model. Below, we outline the aggregation strategies we explored:

**Attention:** having learnable $Q, K, V \in \mathbb{R}^{D' \times D'}$ and $\mathrm{attn}_{h,w} = \mathrm{softmax}_{h,w}(Q\mathbf{e}_{t,1,1}^v, K\mathbf{e}_{t,h,w}^v)$ we compute $\mathbf{z}_t^v = \frac{1}{\sqrt{D'}} \sum_{h=1}^{H'} \sum_{w=1}^{W'} \mathrm{attn}_{h,w} \ V\mathbf{e}_{t,h,w}^v$;

**Summation:** $\mathbf{z}_t^v = \sum_{h=1}^{H'} \sum_{w=1}^{W'} \mathbf{e}_{t,h,w}^v$;

**Mean pooling:** $\mathbf{z}_t^v = \frac{1}{H'W'} \sum_{h=1}^{H'} \sum_{w=1}^{W'} \mathbf{e}_{t,h,w}^v$;

**Max pooling:** $\mathbf{z}_t^v = \max_{(h,w)} \mathbf{e}_{t,h,w}^v$;

**Stacking:** stack embeddings and then project it via a learnable linear layer $\mathbf{L}^v(\cdot) : \mathbb{R}^{H'W'D'} \to \mathbb{R}^{D'}$, $\mathbf{z}_t^v = \mathbf{L}^v([\mathbf{e}_{t,1,1}^v, \mathbf{e}_{t,1,2}^v, \ldots, \mathbf{e}_{t,1,W'}^v, \mathbf{e}_{t,2,1}^v, \ldots, \mathbf{e}_{t,H',W'}^v])$.

We choose VQ-VAE over discriminative video representations (e.g., CLIP, AV-HuBERT, VideoMAE) because VTTS requires preserving fine-grained spatial and temporal facial dynamics rather than global semantic alignment. Discriminative encoders are optimized for semantic matching or classification objectives and often discard spatial locality, whereas VQ-VAE retains a grid of discrete visual tokens encoding mouth shape, jaw motion, and facial muscle activations. These cues are critical for modeling phoneme timing, articulation strength, and emotional intensity. While we do not explicitly supervise emotion, our human evaluation results (Table 2) indicate that these visual tokens implicitly encode facial expressions relevant to speech prosody and emotional delivery.

## F   Evaluation Metrics

**Word Error Rate (WER)**  We use Whisper-large v2 via open-source code https://github.com/m-bain/whisperX to transcribe generated speech. The latter is compared to the ground truth transcription (PL.v2 is treated as a ground truth for VoxCeleb2) to compute WER.

**Mean Opinion Score (MOS)** We use crowd-sourcing to collect subjective ratings to evaluate the intelligibility, naturalness and synchronization of the generated speech. We use the same (randomly sampled) 50 videos from the test set of VoxCeleb2 (or LRS3)[2] for each model to generate speech. We then collect around seven ratings per video for each model. Overall, for both VoxCeleb2 and LRS3, we collect 4208 ratings from 387 different raters. The raters were English-speaking and were paid at least the minimum wage.

We present the raters with a generated speech (with volume normalization) overlayed with the original video or original video with original (or reconstructed) speech. We instruct raters to rate how natural speech in the

---

[2]Speakers in the test sets do not overlap with the speakers from the training sets.

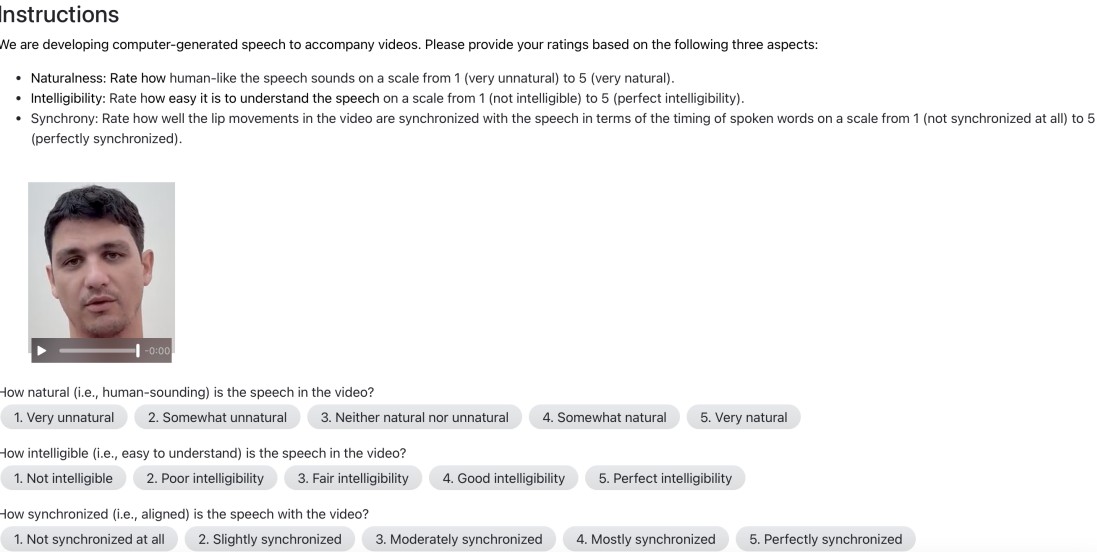

Figure 9: **Human evaluation.** Task description for the crowd-sourced raters to evaluate intelligibility, naturalness and synchronization of the ground truth or generated speech: speech is overlaid with the video and they are played together for the raters.

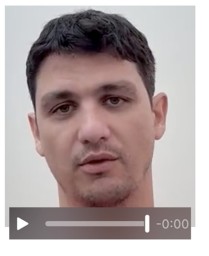

**Instructions**

We are developing computer-generated speech to accompany videos. Your task is to rate how closely the emotions conveyed in the speech match with the emotions displayed through the facial expressions. You should not judge the content or the quality of the speech; rather, just the style and emotions.

How well do the emotions conveyed in the speech match with the emotions displayed through the facial expressions?

1. No match    2. Poor match    3. Fair match    4. Good match    5. Excellent match

Figure 10: **Human evaluation.** Task description for the crowd-sourced raters to evaluate correspondence between facial expressions and emotions in speech for ground truth and generated speech: speech is overlaid with the video and they are played together for the raters.

**Instructions**

Please listen to the two speech samples and rate how similar their emotions are. Focus on how the samples compare in terms of speaking style and emotional tone. Do not consider the content of the speech or the speaker's identity, gender, or audio quality.

| Sample 1 | Sample 2 |
|---|---|

How similar are the emotions in the two speech samples?

1. No similar at all    2. Slightly similar    3. Moderately similar    4. Very similar    5. Extremely similar

Figure 11: **Human evaluation.** Task description for the crowd-sourced raters to evaluate how close emotions in generated speech follows the ground truth.

Table 8: **Human evaluation on VoxCeleb2.** Mean Opinion Scores (MOS) (1-5) with 95% confidence intervals for Intelligibility, Naturalness, and Synchronization. VTTS (VT-Scaled) achieves the best performance in Intelligibility (3.48) and Naturalness (3.20), while VTTS (TV-CoTemporal) performs best in Synchronization (2.50). Both outperform the TTS baseline across all metrics. Ground truth (GT) serves as a reference, and GT (discrete) is an upper bound due to speech discretization.

| Method | Intelligibility ($\uparrow$) | Naturalness ($\uparrow$) | Synchronization ($\uparrow$) |
|---|---|---|---|
| GT | 4.55 $\pm$0.09 | 4.79 $\pm$0.05 | 4.57 $\pm$0.10 |
| GT (discrete) | 3.95 $\pm$0.13 | 3.77 $\pm$0.15 | 4.36 $\pm$0.12 |
| TTS (dmel Bai et al. (2024)) | 3.17 $\pm$0.19 | 2.92 $\pm$0.21 | 1.98 $\pm$0.15 |
| *VTTS* | | | |
| Video-Causal-Streaming | 3.19 $\pm$0.17 | 2.99 $\pm$0.16 | 2.28 $\pm$0.17 |
| TV-CoTemporal | 3.35 $\pm$0.17 | 3.02 $\pm$0.19 | **2.50 $\pm$0.21** |
| VT-Scaled | **3.48 $\pm$0.15** | **3.20 $\pm$0.19** | 2.48 $\pm$0.19 |

video sounds, how intelligible (e.g. easy to understand) speech is in the video, and how synchronized the speech is with the video on a five-point Likert scale, where 1 corresponds to very unnatural and 5 corresponds to very natural. In Figure 9 we show a screenshot seen by raters. Finally, we compute the MOS with confidence intervals calculated using bootstrap resampling with 10k iterations, providing a reliable estimate of the variability MOS results.

We further instruct raters to evaluate emotional consistency between video and generated speech ('video-speech emotions') and emotional expressiveness in speech ('speech emotions') by comparing ground truth and generated audios, see instructions in Figures 10 and 11. MOS results, Table 2, highlight the advantages of visual conditioning for intelligibility, naturalness, synchronization, and emotional expressiveness. We use *emotions2vec*[3] to select audio with non-neutral emotions (6 classes, each prob. $> 99\%$).

**MCD-DTW-SL vs. `TimeSync`**    MCD-DTW-SL measures spectral similarity by computing mel-cepstral distortion (MCD) after dynamic time warping between generated and reference speech. While effective for assessing acoustic fidelity, it operates at the frame level and allows non-linear temporal warping, which can mask synchronization errors by artificially aligning mis-timed speech. In contrast, `TimeSync` operates at the phoneme level and explicitly measures absolute temporal deviation between corresponding phoneme centers without allowing time warping. As a result, `TimeSync` directly captures perceptually salient audio-visual misalignment that DTW-based metrics may obscure, making it better suited for evaluating synchronization in video-conditioned speech generation.

## G    Implementation Details

**Datasets**    1) **LRS3** Afouras et al. (2018) is audio-visual dataset in English, compiled from TED and TEDx video presentations. This dataset stands out for its focus on unconstrained long sentences, featuring a rich vocabulary of over 50k words and thousands of unique speakers. It contains approximately 151k videos with around 439h of speech with transcription. There are 1,452 videos in the test split. 2) **VoxCeleb2** Chung et al. (2018) is a large-scale audio-visual dataset primarily designed for speaker recognition task but applicable to various audio-visual processing domains. It consists of over 1M face-cropped YouTube videos from more than 6k distinct identities, resulting in 1.6k hours of speech *w/o paired transcription.* The dataset is characterized by high variability in lighting conditions, image quality, pose, and motion blur, with an average video duration of 8s. This diversity in real-world conditions makes VoxCeleb2 particularly useful for developing robust models capable of performing well in unconstrained environments. Original data has video at 25fps (40ms per frame), or 25Hz, which we use for video representation extraction, while the audio is given at 16kHz and we extract speech representations at 40Hz (25ms per frame). speakers in train and test data do not intersect.

---

[3]https://huggingface.co/emotion2vec

Table 9: **LRS3 transfer-focused setting.** This setting is introduced in this work for detailed zero-shot transfer analysis. Unlike the main full-set WER results, we additionally report `TimeSync` in this setting.

| Method | Training Dataset | WER (%) ↓ | TimeSync (s) ↓ |
|---|---|---|---|
| V2SFlow (Choi et al., 2025) | LRS3 | 28.5 | - |
| LipVoicer (Yemini et al., 2024) | LRS3 | 21.4 | - |
| Lip2Speech (Kim et al., 2023) | LRS3 | 57.4 | - |
| SVTS (Mira et al., 2022) | LRS3 | 82.4 | - |
| VCA-GAN (Kim et al., 2021) | LRS3 | 90.6 | - |
| DiffV2S (Choi et al., 2023) | LRS3 | 39.2 | - |
| VoiceCraft-Dub* (pretrained) (Sung-Bin et al., 2025) | LRS3 fine-tuned | 1.38 | - |
| TTS | VoxCeleb2 | 5.3 | 0.34±0.28 |
| *VTTS* | | | |
| TV-Global | VoxCeleb2 | 9.5 | 0.24±0.21 |
| VT-Scaled | VoxCeleb2 | 8.2 | 0.32±0.32 |
| VT-Global | VoxCeleb2 | 6.5 | 0.25±0.21 |
| TV-CoTemporal | VoxCeleb2 | **4.8** | **0.22±0.22** |
| VT-Scaled | LRS3 | 6.3 | 0.23±0.24 |
| TV-CoTemporal | LRS3 | 5.6 | 0.23±0.22 |

**Hyper-Parameters Tuning**  To select the best hyper-parameters we randomly sampled 2k samples from the training data and use them as the validation data throughout the training. After we find best hyper-parameters on the validation data, we retrain the final models including validation data into training data.

**Model Training**  For our VTTS models we stack together speaker embedding, video, text and speech representations. Every modality has prepended begin of sentence representation ($< bos >$) and appended end of sentence representation ($< eos >$). Each modality's discrete values are mapped to a common dimension $D'$ through their respective embedding layers and, optionally, additional linear projections before being fed to the decoder. All our models have ∼250M parameters, with $D' = 768$ and 36 transformer layers, following the Base architecture from Bai et al. (2024). We follow masking strategy reported in Bai et al. (2024): for every training step with probability $p$ the sample in the minibatch is masked with the mean span of 3 tokens with masking ratio of 0.5.

We train final models using the AdamW optimizer with a learning rate of $4e − 4$, learning rate warmup of 5k steps, cosine learning rate schedule and gradient clipping of 1.0. We use dynamic batching to optimize the data packing with total batch size of 16.66 minutes. We train all models till full convergence, with 3M maximum number of steps and with mixed precision training (BF16) on H100 GPUs with 80GB. All models are trained with 8GPUs for 3-5 days.

## H   Additional LRS3 Evaluation Setting

The main paper reports primary cross-dataset comparisons on the full LRS3 test set (Table 3) for fairness. In addition, we introduce a transfer-focused setting that keeps clips with durations between 3 and 45 seconds. We choose this range for two practical reasons: (i) clips shorter than 3s tends to have high ASR WER due to ASR sensitivity to shorter audio as the lexical context is limited and ASR (whisper) is trained on 30s audio, and (ii) clips longer than 45s accumulate more ASR/forced-alignment drift, which can dominate synchronization metrics. The same duration filter is applied uniformly to all our models in Table 9.

## I   External-Protocol VoxCeleb2 WER Reference

For completeness, we report published VoxCeleb2 WER numbers from VDTTS (Hassid et al., 2022) as reference-only values. These are not directly comparable to our Table 4 because the evaluation protocol differs in data preprocessing/filtering, ASR backend, and transcript pipeline.

Table 10: **Reference-only VoxCeleb2-trained WER values.** All rows are trained on VoxCeleb2 and evaluated on VoxCeleb2. Rows from (Hassid et al., 2022) use an external evaluation protocol; our best VoxCeleb2-trained model is included for context from Table 4. These rows are not directly comparable.

| Method | WER (%) ↓ |
|---|---|
| VDTTS-VOXCELEB2 (Hassid et al., 2022) | 48.0 |
| VDTTS-LSVSR (Hassid et al., 2022) | 25.0 |
| VTTS (VT-Scaled, VoxCeleb2-trained, ours) | 12.2 |

## J  Video-to-Speech

As one of the baselines we trained speech generation model conditioning only on the video input (no text input). The WER for this model is around 100%, and MOS is $1.39 \pm 0.10$ for intelligibility, $1.60 \pm 0.13$ for naturalness and $1.49 \pm 0.09$ for synchronization. The interesting findings about this model are: a) the model is able to generate word $n$grams; b) the model is able to model properly the pauses and reflect the timing when people speaking or being silent.

## K  Qualitative Results

In Figures 12, 14, and 16, we show log mel-spectrogram comparisons between TTS, Ground Truth (GT), and our VTTS (VT-Scaled) model across different scenarios. These visualizations include both successful cases where VTTS (VT-Scaled) effectively captures temporal dynamics and spectral patterns, and failure cases (Figure 16) that highlight current limitations. Through these examples, we can analyze how video conditioning helps maintain proper speech duration and temporal alignment, while also identifying challenges in generating complex spectral information. Furthermore, to analyze temporal synchronization between generated and ground truth speech, we visualize phoneme-level alignments in Figures 13, 15, and 17. Each plot shows the relationship between phoneme timings in ground truth ($x$-axis) versus generated speech ($y$-axis), where perfect synchronization would follow the diagonal dashed line. The different variants of our model, VTTS (VT-Scaled) consistently demonstrate better temporal alignment compared to TTS, as evidenced by their closer distance to the ideal diagonal. This visualization helps quantify how video conditioning helps to maintain proper speech timing and rhythm, with VTTS (VT-Scaled) variants showing improved temporal coherence across different examples.

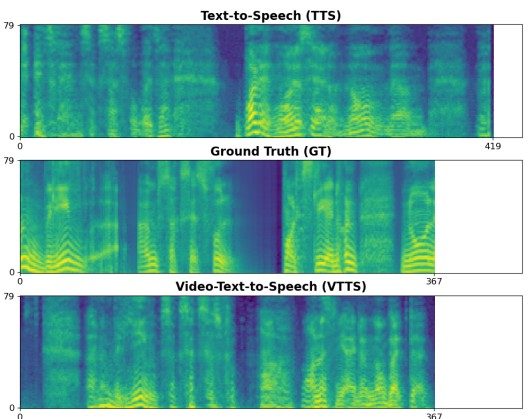

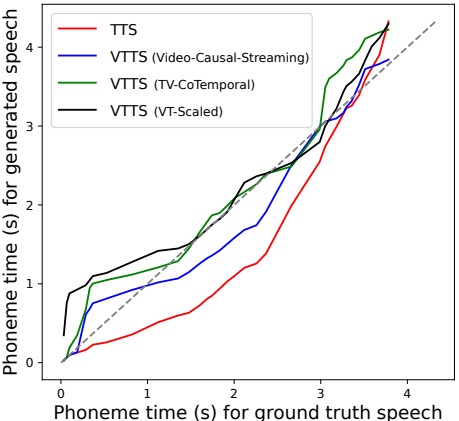

Figure 12: **Qualitative comparison of log mel-spectrograms.** Visualization of generated log mel-spectrograms: Text-to-Speech (TTS, top), Ground Truth (GT, middle), and our Video-Text-to-Speech (VTTS, bottom). VTTS (VT-Scaled) demonstrates better temporal alignment with GT (367 frames) compared to TTS (419 frames), showing the benefit of video conditioning for maintaining correct speech duration. The spectral patterns in VTTS (VT-Scaled) also more closely match GT's energy distribution.

Figure 13: **Alignment between phonemes.** Temporal alignment visualization for example from Figure 12. The plot compares phoneme timings between ground truth ($x$-axis) and generated speech ($y$-axis). Dashed gray line is the ideal time synchronization between GT and generated speech. TTS is way out for the proper timing compared to VTTS (TV-CoTemporal) and VTTS (VT-Scaled).

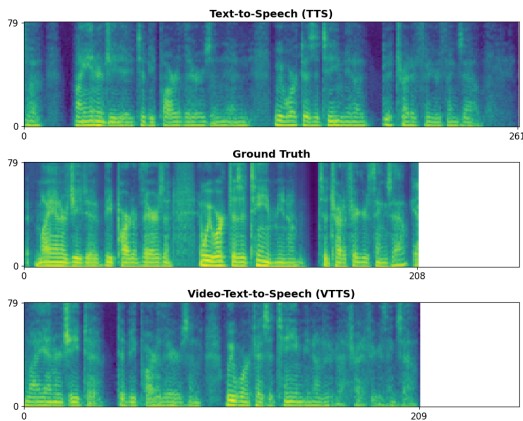

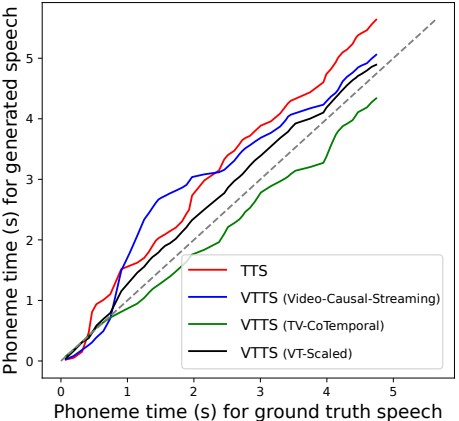

Figure 14: **Qualitative comparison of log mel-spectrograms.** Visualization of generated log mel-spectrograms from different methods: Text-to-Speech (TTS, top), Ground Truth (GT, middle), and our Video-Text-to-Speech (VTTS, bottom). VTTS (VT-Scaled) demonstrates better temporal alignment with GT (208 frames) compared to TTS (261 frames), showing the benefit of video conditioning for maintaining correct speech duration. The spectral patterns in VTTS (VT-Scaled) closely match GT's harmonic structure and energy distribution, particularly visible in the lower frequency bands (yellow regions). Additionally, VTTS (VT-Scaled) accurately captures the temporal dynamics of speech, including pauses and intensity variations, leading to more natural speech generation.

Figure 15: **Alignment between phonemes.** Temporal alignment visualization for example from Figure 14. The plot compares phoneme timings between ground truth ($x$-axis) and generated speech ($y$-axis). VTTS demonstrate superior temporal alignment by following the ideal synchronization line (dashed diagonal) more closely than TTS, which shows significant temporal drift. This example highlights how video conditioning helps maintain proper speech timing.

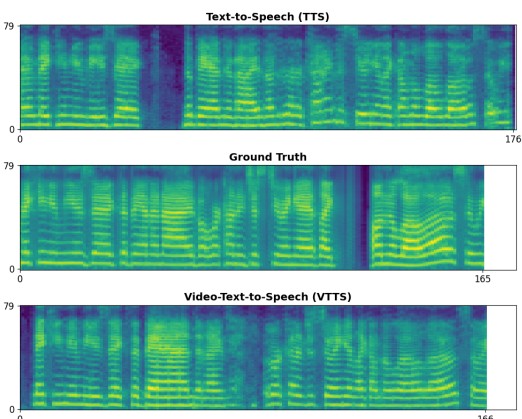

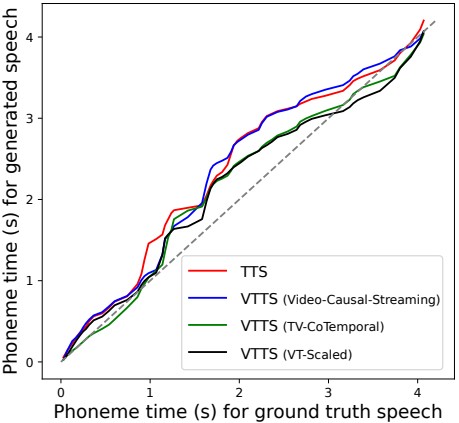

Figure 16: **Failure case analysis of log mel-spectrograms.** Visualization of generated log mel-spectrograms from different methods: Text-to-Speech (TTS, top), Ground Truth (GT, middle), and our Video-Text-to-Speech (VTTS, bottom). While VTTS (VT-Scaled) maintains better temporal alignment with GT (165 frames vs TTS's 176 frames), both VTTS (VT-Scaled) and TTS struggle to accurately capture GT's harmonic structure and energy distribution. Despite having video conditioning, VTTS (VT-Scaled) shows degraded spectral quality particularly in the mid-frequency ranges, though it still preserves some temporal speech dynamics like pauses. This example highlights current limitations in generating complex spectral patterns.

Figure 17: **Alignment between phonemes.** Temporal alignment visualization for failure case corresponding to Figure 16. Plot shows phoneme timing comparison between ground truth ($x$-axis) and generated speech ($y$-axis). While VTTS maintain better alignment than TTS, all models show deviation from ideal synchronization (dashed diagonal), particularly in later segments, illustrating challenges in maintaining temporal coherence for complex speech patterns.

