# OpenReview forum: "Mechanisms of Multimodal Synchronization: Insights from Decoder-Based Video-Text-to-Speech Synthesis"
_TMLR — Under review for TMLR_

### Review · Reviewer_fVtw · 2026-06-13

**Summary Of Contributions:**

This paper studies how unified decoder-only multimodal transformers synchronize modalities with heterogeneous sampling rates, using video-text-to-speech (VTTS) synthesis as a controlled testbed. The authors introduce Visatronic, a unified decoder-only model that represents speaker, video, text, and speech as discrete tokens in a shared sequence. The paper investigates how modality contributions, position-ID strategies, and modality ordering affect synchronization and generalization. It also proposes TimeSync, a phoneme-level alignment metric based on forced alignment, intended to diagnose timing errors that are not localized by common frame-level metrics.

The main strengths are: (1) the problem is timely and relevant to unified multimodal generation; (2) the paper studies synchronization mechanisms through controlled architectural/sequence-design ablations rather than only reporting end-task performance; (3) the empirical evaluation is fairly broad, including WER, TimeSync, MOS, emotional expressiveness, in-domain results, and zero-shot transfer; and (4) the paper is generally careful in distinguishing directly comparable baselines from reference-only systems.

The main weaknesses are: (1) some claims about “mechanisms” and transferability are stronger than what the evidence fully establishes, since the study is limited to one main task family, two datasets, and models trained from scratch at moderate scale; (2) TimeSync is useful but would benefit from stronger validation against human synchronization judgments and existing metrics; (3) several comparisons are not fully controlled because baselines differ in task setting, input modality, training data, and pretraining; and (4) the paper would benefit from clearer reporting of statistical significance, failure modes, and the robustness of conclusions across random seeds and hyperparameter choices.

**Additional Comments:**

I enjoyed reading the paper. The core idea of using VTTS as a controlled testbed for studying multimodal synchronization is compelling, and the ablation-driven approach is valuable. The paper is strongest when it focuses on concrete design insights: text and video provide complementary signals; ordering and position-ID assignment matter; and phoneme-level timing metrics can reveal synchronization failures that aggregate metrics miss.

My main concern is that the paper occasionally overgeneralizes from a specific experimental setup. With more careful wording, stronger TimeSync validation, and clearer baseline framing, this would be a useful contribution to TMLR.

**Audience:**

Yes

**Audience Explanation:**

The paper should be of interest to researchers working on multimodal learning, speech synthesis, video-conditioned generation, temporal alignment, and unified decoder-only architectures. The question of how decoder-only models handle heterogeneous sampling rates is important and underexplored. The paper’s controlled comparison of position-ID assignment and modality ordering provides useful empirical guidance for future multimodal sequence design. The proposed TimeSync metric is also potentially useful as a diagnostic tool for audio-visual synchronization, even if it requires further validation.

The work is also relevant beyond VTTS, because many multimodal generation tasks require aligning sparse symbolic inputs, dense visual streams, and continuous audio or motion outputs. That said, the paper’s broader relevance would be stronger if the authors more clearly stated which findings are expected to transfer beyond VTTS and which are specific to the chosen speech/video discretization pipeline.

**Broader Impact Concerns:**

The paper studies video-conditioned speech generation, which has clear dual-use risks. The main concerns are impersonation, voice spoofing, unauthorized dubbing, and generation of misleading audio-visual content. The paper includes an ethics discussion that mentions dual-use risks, privacy, accessibility, transparency, and the decision not to release pretrained models. This is a good start.

However, the broader impact discussion should be made more concrete. The authors should describe what safeguards are recommended in practice, such as watermarking, consent requirements, dataset governance, detection protocols, and restrictions on deployment. They should also discuss whether the use of VoxCeleb2 and public videos raises consent concerns for speech/face synthesis, even if the dataset is publicly available for research. I do not see these concerns as blocking acceptance, but they should be addressed more explicitly before publication.

**Claims And Evidence:**

Yes

**Claims Explanation:**

Overall, the central empirical claims are supported by a substantial set of experiments. The ablation studies show that both text and video are important: removing either modality causes a large degradation in WER. The position-ID and ordering comparisons provide evidence that synchronization quality and cross-domain transfer depend on sequence design. The VoxCeleb2 and LRS3 results also support the claim that the proposed text-first/co-temporal ordering is more robust for transfer, while the video-first variant is stronger in-domain. The human evaluations further support the claim that visual conditioning can improve perceived synchronization and emotional consistency.

However, several claims should be softened or better qualified. First, the paper sometimes frames the findings as revealing general “mechanisms” of multimodal synchronization in decoder-only models, but the evidence is limited to VTTS, to specific tokenization/discretization choices, and to a model scale below large modern multimodal systems. The work is best interpreted as a controlled empirical study of synchronization behavior under these design choices, rather than a general explanation of synchronization in all unified decoders.

Second, TimeSync is a promising diagnostic metric, but the paper does not yet provide enough evidence that it correlates reliably with human perception across conditions. It is also sensitive to forced-alignment quality, ASR/transcript quality, and the choice to remove silence segments. These design decisions are reasonable, but they should be discussed more explicitly and validated with correlation or sensitivity analysis.

Third, some baseline comparisons need clearer framing. The authors do state that some methods are not directly comparable, but the paper should more consistently separate mechanism-oriented comparisons from performance comparisons against specialized or pretrained systems. The strong VoiceCraft-Dub result, for example, is correctly marked as reference-only, but its presence may still invite overinterpretation.

**Requested Changes:**

Critical to acceptance:

1. Please temper the “mechanisms” language throughout the paper. The experiments provide useful empirical evidence about the effects of ordering and position-ID assignment in this VTTS setup, but they do not fully establish general mechanisms for all unified multimodal decoders. I recommend reframing the main claims as “empirical evidence” or “design insights” unless supported by additional analysis.
2. Please strengthen the validation of TimeSync. At minimum, the paper should discuss its sensitivity to forced-alignment errors, transcript/ASR errors, silence removal, and phoneme substitution alignment. Ideally, the authors should report correlation between TimeSync and human synchronization MOS, or provide an analysis showing when TimeSync agrees or disagrees with human judgments and frame-level metrics.
3. Please clarify comparability of baselines more consistently. Some comparisons involve different input modalities, training datasets, pretrained components, or fine-tuning setups. The current text partially acknowledges this, but the paper should more clearly distinguish: (i) mechanism/ablation comparisons within the proposed framework, (ii) directly comparable prior work, and (iii) reference-only systems.
4. Please add more statistical evidence for the key ordering and position-ID conclusions. Some differences are small, especially among the in-domain WER and TimeSync values. Confidence intervals, multiple seeds, or at least a discussion of expected variance would make the conclusions more convincing.
5. Please clarify the role of pseudo-label quality in VoxCeleb2. Since PL.v2 transcripts are treated as ground truth for WER and alignment-related analyses, the paper should better quantify pseudo-label noise and discuss how it affects WER, TimeSync, and the interpretation of model improvements.

Would strengthen the work:

1. Include a clearer table summarizing all sequence variants, their token order, position-ID assignment, causal constraints, and intended use case. This would make the paper easier to follow.
2. Add more failure-case analysis in the main paper, not only the appendix. The appendix failure cases are useful, especially cases where temporal alignment improves but spectral quality remains poor.
3. Provide more detail on human evaluation: rater screening, number of ratings per condition, whether raters saw randomized/blinded samples, and how confidence intervals were computed.
4. Discuss the practical implications of not releasing pretrained models. This is understandable for safety/privacy reasons, but it affects reproducibility. Releasing code, data processing scripts, pseudo-labels, and evaluation scripts becomes especially important.
5. Improve writing clarity and grammar in several places. Examples include awkward phrasing such as “talking people,” minor punctuation issues, and occasional overstatements in the abstract and introduction.

As a minor presentation/privacy concern, Figure 2 appears to contain an identifiable human face in the example video frame. I recommend anonymizing or blurring this face, unless the authors can confirm that appropriate consent and rights have been obtained for displaying it.

---

### Review · Reviewer_QWAw · 2026-06-20

**Summary Of Contributions:**

Authors pursue the task of VTTS to understand synchronization of 3 modalities (video, audio, text). They train a decoder using pretrained models for encoding input into discrete tokens. Authors try global and matched positional encoding primarily to comment which works better for metrics of interest such as WER and phoneme alignment for which they also propose TimeSync metric. Through cross-domain evaluation and human MOS, authors prescribe which positional encoding scheme is suited better.

**Audience:**

Yes

**Audience Explanation:**

Video analysis, TTS, and synchronization are suitable topics.

**Broader Impact Concerns:**

Nil

**Claims And Evidence:**

No

**Claims Explanation:**

1. Using a single training set and two test sets (one in-domain, one out-domain) is insufficient for answering the questions authors set out to answer. The training set is also very small. The conclusion is also divided - no scheme is overall best. Simplest thing author could have switched data A (voxceleb) and data B (lrs3) and repeated the analysis. (It is ok if conclusion remains same after this experiment.)
2. The core of proposal is quite short compared to total manuscript length. I was expecting more schemes to try since novelty is very limited in the paper (which is ok). A very simple scheme could be concatenation of video and text (with appropriate repeating). Isn't this the simplest "synchronization" mechanism? If this gives good results, it will require further justification for the proposed schemes!
3. By removing silence from TimeSync, i am not sure if the synchronization analysis is complete. or is it the silence handling can be easily addressed in some post-processing? Authors could have reported TimeSync-raw which has silence accounted for. The output audio must match with the video lip movement. I believe the metric dismisses substitution errors? Atleast the average could be weighted average (by phoneme frequency) in my opinion.
4. Authors state they remove all confounding factor in modern systems so they can study synchronization via VTTS task.  But their quantization based modeling has caused 10.1% WER alone! no TTS/VTTS is involved yet and we are starting at a bad goal to touch. GT WER is around 2% only. So their "controlled playground" is less convincing framing. Or atleast, we can report different baseline TTS systems so general conclusions can be drawn. Also, why focus on zero-shot evaluation? We can always tune a bit and see if observations change. (This is also a sanity check experiment)
5. Fig7 shows an example where i see green line below the dotted line. This is good for illustration but is this a general trend? I was expecting some statistics to summarize these plots

**Requested Changes:**

All experiments mentioned above, potentially with more datasets, more positional embeddings, more analysis into timesync variants, more TTS variants, LSE analysis, impact of noise in data (into both modalities). This is important to make general conclusions into the preliminary observations authors report (such as Section 4.3 last line). Also, better baselines are required.

---

### Review · Reviewer_NE6i · 2026-07-21

**Summary Of Contributions:**

Although a lot is known about multimodal learning architectures, especially in the context of audio-visual-speech/text learning; we do not have clarity on the finer details of token level and temporal (time) stamp level alignment and ordering needed when it comes to designing efficient generalizable multimodal architectures. Clearly transformers with their token ordering comes to mind as the key architecture to tease-apart such relations. The authors address this problem.

They choose to give away with encoder-decoder transformer designs and instead focus on decoder only setups with a unified token representation / ordering wherein there is a way to explicitly hand-code/manipulate the text, video, speech and speaker/id tokenization, along with parameterizing the time alignment (time sync of the individual signal chunks) of the unimodal signals. This essentially provides a nice framework to tinker with various combinations of token ordering for each modality and time sync windows; and the decoder only transformer can then be used to benchmark the setup. The framework is flexible enough to ablate various combinations of ordering and synchronization.

The proposed framework is thoroughly evaluated using the traditional AV datasets like LRS3 and voxceleb2. In addition to quantitative evaluations in speech signal recovery from video-text, the authors also use human evaluations on realism (quality, intelligibility) of generated speech - this is to ensure the analysis is anchored on valid generation mechanisms.

**Additional Comments:**

None

**Audience:**

Yes

**Audience Explanation:**

The analysis framework is quite general and can be adopted to any multimodal designs / datasets. Even if the explicit findings may not be applicable, the procedure can be replicated.
Voxceleb2 is a well studied datasets in the content of speech/audio synthesis and the design outcomes from CoTemporal and VT-scaled can influence modeling choices in time-domain speech generation from AV content and emotional speech design studies.

**Broader Impact Concerns:**

Better audio/speech and visual synchronization in data generation comes with the risk of making deep fakes more realistic and plausible. Some commentary on how the findings may help improve such deepfakes and what can be done to mitigate that scenario is needed.

**Claims And Evidence:**

Yes

**Claims Explanation:**

The proposed framework of decoder only setting which unifies individual tokenization setups and allows time synchronization to be setup at the unified/joint representation level is quite simple by design -- and hence, its quite interpretable.
CoTemporal designs turned out to be best, and this is generally also observed in traditional multimodal learning where high quality video with tags (low sample rate text) captures most of the information about speech.
Although the choices are datasets are limited, the evaluation in general was rigorous; justifying the claims about generality of the findings.
Human evaluation studies showed 'expected' trends that align with the signal recovery outcomes - justifying the correctness of decoder only architectures in modeling the multimodal content i.e., the chosen architectures are not too naive to begin with.

**Requested Changes:**

Major points:
1. What is the sensitivity of the orderings and synchronization? i.e., how sensitive are the findings/outcomes to noise in inputs (text, video) and noise in speech (e.g., background noise)? In theory one should expect subtle changes to token ordering across T-V-S to not effect the underlying decoder output but that is nevertheless a hypothesis that is to be tested in the context of this framework.
2. Consider the 3 canonical orderings used - tv-cotemporal, video-causal, vt-scaled - could we have parameterized the orderings and the position ids + time stamps in these in the sense that a specific parameterization can 'produce' one of the 3 canonical orderings? In that same sense, can such a parameterization produce other non-canonical orderings e.g., something between co-temporal and vt-scaled where one half of the signal in a time chunk follows one ordering and the other half uses another ordering? This question is coming from the place where we have a better way to formulate and parameterize this framework the authors are proposing. Maybe I am missing something, could you clarify?
2.1 A related question is -- why choose these specific ordering? In essence reversing the speech and video-text ordering could also generate a cotemporal and vt-scaled orderings that can still be used for speech generation isn't it?
3. The presentation / description of content in the section 4 is quite hard to follow -- any attempt to make it more concise will help with the overall flow of the paper.

Minor points:
1. Figure 4 and 6 are quite difficult to follow-up. Is there a better way to describe what is happening with re: to the position ids, the token ranks/order and the modality sequencing?
2. How do we interpret the y-axis in Figure 8?
3. Table 6 is not saying much, as in, the nature of these datasets is that each modality is in fact presenting some non trivial signal and so missing/dropping modalities will of course reduce performance. It's not necessarily adding value to the technical content.
4. Is the Timesync calculated over the entire avt sequence? That seems too long for phoneme level matching/sync isn't it? Is there a master time window within which this is calcuated? What would be a bad offset look like (and I am guessing this is dataset dependent)?
5. Are the human raters / evaluators experts in speech/audio domain? Were the evaluations A2b? And were there x-signal evaluations where one sample is reviewed by multiple raters?

---

### Author Response · Authors · 2026-06-10
**Review Status for our submission**

I am writing to kindly inquire about the review status of our submission. We would be grateful to know whether any reviewers have been assigned to the paper yet, and whether there is any update regarding the expected review timeline.
Thank you very much for your time and for handling our submission. We greatly appreciate your help.

---

> ### Comment · Action_Editor_qu5w · 2026-06-10
> **Reviewers have been assigned**
>
> This paper has been assigned 3 reviewers, with reviews due by the end of the day on Friday, June 12.
>
> AE